# RETHINKING ACTOR-CRITIC: SUCCESSIVE ACTORS FOR CRITIC MAXIMIZATION

## ABSTRACT

Value-based actor-critic approaches have been widely employed for continuous and large discrete action space reinforcement learning tasks. Traditionally, an actor-network is trained to find the action that maximizes the critic (action-value function) with gradient ascent. We identify that often an actor fails to maximize the critic because (i) certain tasks have challenging action-value landscapes with several local optima, and (ii) the critic landscape varies non-stationarily over training. This inability to find the optimal action often leads to sample-inefficient training and suboptimal convergence. To address the challenge of better maximization of the critic's landscape, we present a novel reformulation of the actor by employing a sequence of sub-actors with increasingly tractable action-value landscapes. In large discrete and continuous action space tasks, we demonstrate that our approach finds actions that better maximize the action-value function than conventional actor-network approaches, enabling better performance. https://sites.google.com/view/complexaction

## 1 INTRODUCTION

Reinforcement Learning (RL) has increasingly been employed for solving complex decision-making problems, particularly through off-policy algorithms that learn an action value or Q-function for better sample utilization. Algorithms like Q-learning (Mnih et al., 2015) excel in small, enumerable action spaces by exhaustively evaluating Q-values to select the best action. However, this is computationally impractical for tasks with large-discrete or continuous action spaces, like robotics and recommender systems. Therefore, actor-critic approaches have emerged, wherein a policy (actor) is trained to select the action that maximizes the learned value function (or critic).

In this work, we identify a key practical drawback in actor-critic methods: an actor often fails to find actions that truly maximize the critic. The core issue stems from the use of gradient ascent over the action space, where the optimization objective is the critic's value output. This is commonly employed in algorithms like DDPG (Lillicrap et al., 2015), TD3 (Fujimoto et al., 2018), and SAC (Haarnoja et al., 2018). However, this action-value optimization landscape is often non-convex and high-dimensional, and changes non-stationarily during training. Consequently, when the actor does not sufficiently maximize the critic, it results in low-reward ineffective data for training, inaccurate Bellman updates for the critic (Sutton and Barto, 2018), and crucially, a suboptimal final agent.

Our goal is to address the challenge of finding the action that maximizes the critic in complex landscapes. Our key hypothesis is that an ensemble of successive actor-critic modules can collectively globally optimize the Q-value landscape than a single, monolithic actor. Thus, we introduce a sequence of additional actor-critic modules that work alongside the primary actor-critic pair. The set of modules provides a set of actions, which are then evaluated with the critic: more actions mean a better likelihood of critic maximization. Importantly, we introduce a novel training objective for each actor-critic module: each module's selected action sets a Q-value baseline over the critic in the subsequent modules. Thus, successive modules have an optimization landscape "pruned" of all actions with Q-values lower than this baseline.

Our approach, named SAVO (Successive Actors for Value Optimization), is particularly beneficial in tasks that feature high-dimensional action spaces or Q-value landscapes with discontinuities and many local optima. Apart from continuous control tasks, another crucial application is in large-discrete

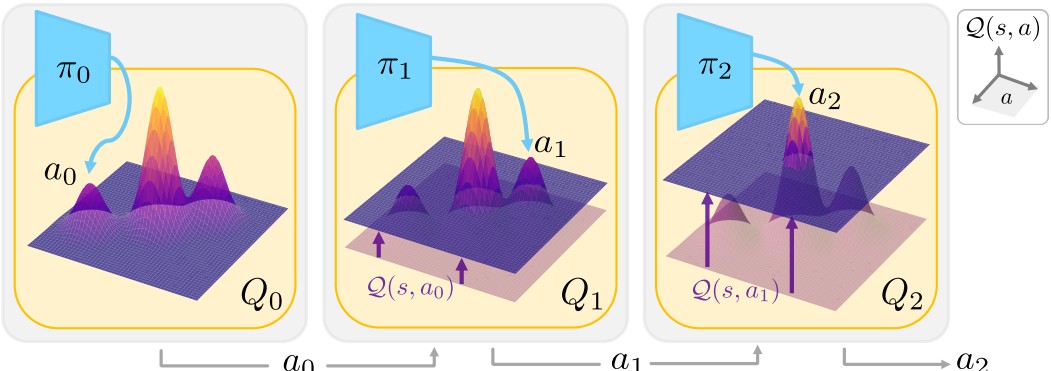

Figure 1: For a given state, the Q-value landscape can be hard to optimize for a single actor due to the presence of local optima. Our approach trains successive actor-critic agents such that the action $a_0$ selected by an actor sets a baseline value of $\mathcal{Q}(s, a_0)$ in the subsequent critic. This prunes away $Q_1$'s landscape below the baseline, thus making the optimization task of $\pi_1$ more tractable. Finally, all selected actions are explicitly evaluated by the primary Q-function to find the most optimal action.

action spaces with associated action representations. The conventional value-based approach (Dulac-Arnold et al., 2015) is an extension of DDPG and thus also suffers from the drawback of insufficient critic optimization. Furthermore, large-discrete action space tasks like recommender systems have an especially complex critic landscape, as the discrete actions only exist in distinct points in this space. For instance, two seemingly similar movie recommendation actions can have sharply different Q-values, thus leading to several discontinuities and local optima. Therefore, we apply our algorithm SAVO as a general solution for tasks with challenging action-value landscapes in both continuous and large-discrete action spaces.

Our key contribution is formulating the challenge of effective optimization of the critic landscape in actor-critic algorithms for RL. We reformulate the actor-critic framework as a sequence of actor-critics trying to successively find an action with a better critic value. We validate our approach through a comprehensive set of experiments, including continuous control tasks with varying optimization difficulties and discrete action space tasks, such as a mining environment and simulated and real-data recommender systems. Our method outperforms traditional actor-critic algorithms and alternative approaches we propose, showing that better maximization of the critic leads to better performance.

## 2 RELATED WORK

### 2.1 OPTIMIZATION OF Q-FUNCTION

Q-learning (Watkins and Dayan, 1992; Tesauro et al., 1995) is a fundamental model-free reinforcement learning algorithm that learns to make optimal decisions by iteratively estimating and updating the Q-values. Deep Q-learning (Mnih et al., 2015) has been applied to tasks with tractable discrete action spaces such as Atari (Mnih et al., 2013; Espeholt et al., 2018; Hessel et al., 2018), traffic control (Abdoos et al., 2011), and small-scale recommender systems (Chen et al., 2019). Q-learning requires a maximization over the action space for two reasons: (i) computing the next state's optimal value to set as a target for the current state's value and (ii) acting in the environment during evaluation and exploration. Extending Q-learning to continuous or large discrete action spaces poses challenges due to the computational infeasibility of enumerating and evaluating all possible actions.

### 2.2 MODIFYING Q-FUNCTION FOR EASIER OPTIMIZATION

Some prior work make Q-function more easily optimizable by training convex quadratic Q-functions by combining Q-learning with a PID controller (Wang et al., 2019) and proposing a continuous variant of the Q-learning algorithm known as Normalized Advantage Functions (Gu et al., 2016). Wire fitting algorithm (Baird and Klopf, 1993) modifies the Q-function representation to be over a set of functions, called control wires, such that the maximization problem is a constant time operation. However, these do not scale to online RL where the Q-function is trained over millions of steps.

### 2.3 ACTOR-CRITIC Q-OPTIMIZATION

**Gradient-Ascent approaches:** Actor-critic approaches differ between on-policy and off-policy reinforcement learning methods. In on-policy methods like REINFORCE (Williams, 1992), TRPO (Schulman et al., 2015) and PPO (Schulman et al., 2017), the actor aims to directly maximize expected returns, while the critic helps reduce variance and improve return estimates. In contrast, in off-policy value-based actor-critic methods, the critic is the primary learner, and the actor follows it with gradient ascent, enabling the search for the best action in continuous action spaces. DPG and DDPG (Silver et al., 2014; Lillicrap et al., 2015) extended these concepts to formulate gradient descent objectives for continuous action spaces, while TD3 (Fujimoto et al., 2018) addressed practical function approximation issues. SAC (Haarnoja et al., 2018), on the other hand, combines the critic's objective with entropy maximization but retains stochastic gradient ascent for actor training.

**Sampling-based approaches:** For large-discrete action space tasks, such as large-scale recommendation systems (Zhao et al., 2018; Zou et al., 2019; Wu et al., 2017), conventional discrete action RL is infeasible as it requires enumeration and evaluation of all actions' values via Q-networks. This is usually dealt with associating the discrete actions with continuous representations and then solving the continuous action space RL problem. Specifically, Wolpertinger (Dulac-Arnold et al., 2015) extended the DDPG algorithm by enabling the actor to act in the space of continuous action representations, followed by a k-NN step, and finally the Q-function is used to evaluate the best action among the k actions. While sampling k-nearest neighbor actions around the actor's output improves the maximization of the critic, it is only a heuristic solution and often requires $k$ to be infeasibly large – up to 10% of the action space in Dulac-Arnold et al. (2015). In experiments (Sec. 6), we show how such sampling-based approaches fail in tasks where the Q-value landscape is challenging.

**Ensemble-based approaches**: Ensembles of policies enhance exploration in reinforcement learning (Osband et al., 2016; Chen and Peng, 2019; Song et al., 2023; Zheng12 et al., 2018; Huang et al., 2017), but do not directly solve the problem of finding better globally optimal actions (Sec. 6).

### 2.4 EVOLUTIONARY ALGORITHMS FOR Q-OPTIMIZATION

Evolutionary algorithms like simulated annealing (Kirkpatrick et al., 1983), genetic algorithms (Srinivas and Patnaik, 1994), tabu search (Glover, 1990), and cross-entropy method (CEM) (De Boer et al., 2005) are often employed in global optimization problems (Hu et al., 2007). Certain evolutionary approaches have focused on improving exploration in RL (Colas et al., 2018; Maheswaranathan et al., 2019; Khadka and Tumer, 2018), but do not focus on better Q-value optimization.

Specifically for Q-value optimization, QT-Opt and its variants (Kalashnikov et al., 2018; Lee et al., 2023; Kalashnikov et al., 2021) utilize CEM as the actor. Hybrid of evolutionary and gradient descent actor-critic approaches also exist. CEM-RL (Pourchot and Sigaud, 2018) improves stability and hyperparameter-sensitivity of TD3 with the help of CEM. GRAC (Shao et al., 2022) which trains the actor with a combination of gradient ascent over Q-function and imitating high-value actions from CEM. Cross-Entropy Guided Policies (Simmons-Edler et al., 2019) trains a policy network to imitate CEM's sampling behavior for faster inference. Algorithms with CEM require infeasible amounts of repetitive evaluations of the Q-function and do not scale to high-dimensional action spaces (Yan et al., 2019). In contrast, SAVO directly tackles the challenge of pruning local optima, and thus performs better than a CEM actor (Section 6). Finally, Greedy Actor-Critic (Neumann et al., 2018) emulates CEM by sampling from an action proposal policy, evaluating on the Q-function, and training the actor to greedily follow top actions. This approach is also prone to local optima due to the dependence on gradient ascent and training a single actor, as we empirically demonstrate in Section 6.

## 3 PROBLEM FORMULATION

In this work, we tackle the problem of how to effectively optimize the action-value landscape in actor-critic methods for both continuous and large-discrete action space tasks. We represent a task with a Markov Decision Process (MDP), defined by a tuple $\{\mathcal{S}, \mathcal{A}, \mathcal{T}, \mathcal{R}, \gamma\}$ of states, actions, transition probability, reward function, and a discount factor, respectively. The action space $\mathcal{A}$ is a $D$-dimensional continuous vector space, $\mathbb{R}^D$. At each time step $t$ in the episode, the agent receives a state observation $s_t \in \mathcal{S}$ from the environment and acts with $a_t \in \mathcal{A}$. Then, it receives the new

state after transition $s_{t+1}$ and a reward $r_t$. The objective of the agent is to learn a policy $\pi(a|s)$ that maximizes the expected discounted reward, $\max_\pi \mathbb{E}_\pi \left[ \sum_t \gamma^{t-1} r_t \right]$.

**Discrete action space**: Given a discrete action set $\mathcal{A}_\alpha$, we define a mapping $\mathcal{R} : \mathcal{A}_\alpha \to \mathcal{A}$, such that $\mathcal{R}(\alpha) = a_\alpha$ is the continuous representation for action $\alpha \in \mathcal{A}_\alpha$. The inverse mapping is defined with a nearest neighbor function, $f_{\text{1-NN}} : \mathcal{A} \to \mathcal{A}_\alpha$, such that $\hat{\alpha} = f_{\text{1-NN}}(a) = \arg\min_{\alpha \in \mathcal{A}_\alpha} \|\mathcal{R}(\alpha) - a\|_2$. For example, in a movie recommender system, the representation function $\mathcal{R}$ may be a combination of the movie's metadata, text embeddings of its reviews, and visual embedding of the movie poster.

### 3.1 DETERMINISTIC POLICY GRADIENTS (DPG)

Silver et al. (2014) proposed the DPG algorithm where an actor $\pi_\phi$ is updated through gradient ascent over the critic, and the critic is learned with the Bellman equation (Bellman, 1966):

$$\nabla_\phi J(\phi) = \mathbb{E}_{s \sim p_\pi} \left[ \nabla_a \mathcal{Q}^\pi(s, a) \big|_{a=\pi(s)} \nabla_\phi \pi_\phi(s) \right] \tag{1}$$

$$\mathcal{Q}_\pi(s, a) = r + \gamma \mathcal{Q}(s', a'), \quad a' \sim \pi_\phi(s'). \tag{2}$$

Follow-up works like DDPG and TD3 addressed the practical issues relating to function approximation with deep neural networks, resulting in a stable algorithm that works well for a large variety of continuous control tasks. Meanwhile, Soft Actor-Critic (SAC) employs a DPG-style policy gradient, but for stochastic policies. In large-discrete action spaces, Dulac-Arnold et al. (2015) extend DDPG with k-nearest-neighbors over the actor's output, $f_{\text{k-NN}}$, and subsequent critic maximization.

However, all of these algorithms and their follow-ups share that the actor is trained with gradient ascent over the action-value or $\mathcal{Q}$ landscape, following Equation 1. To achieve a unified architecture for continuous and large-discrete action spaces, this paper uses TD3 as the base algorithm across our formulation and all experiments. However, the identified challenge and ideas are broadly applicable to any algorithm training actors to optimize the critic with gradient ascent, including SAC.

### 3.2 THE CHALLENGE OF ACTOR OPTIMIZING THE CRITIC

The key challenge for an actor to maximize the action-value optimization landscape is that it is often high-dimensional and non-convex, replete with local optima and discontinuities. For instance, in high-stake precision tasks like surgical robots, actions with high Q-values would lie in a sharp optimal region (Barnoy et al., 2021) between several suboptima. Similarly, for table-top manipulation or locomotion robots like humanoids, there are several regions of the action space that are invalid or unsafe, leading to a discontinuity in the Q-function space (Florence et al., 2022). Further, for large-discrete action space problems like movie recommenders (Zhou et al., 2010), two seemingly similar movies (actions) could be nearby in the representation space of movies, yet may have completely different utilities for a particular user (Jain et al., 2021; 2020). Such Q-function spaces would be plagued with local optima, making it hard for the actor to learn to find the action maximizing the Q-function. These challenges are exacerbated by the Q-value landscape evolving as the critic itself trains. From Eq. 1, we see that actors trained with gradient ascent are only informed in the local neighborhood of their current action. Thus, a suboptimal actor can only gradually follow the updates to the Q-function.

These challenges lead to actions that do not sufficiently optimize the Q-function. This leads to an ineffective agent in three ways. First, the data generated during the agent's training yields low rewards, rendering it ineffective for training. Second, the actor's inability to optimize the Q-function leads to inaccurate and misdirected Bellman updates for the critic (Eq. 2). Third, and most critically, actors stuck in the local optima of the Q-function lead to a suboptimal agent at convergence.

## 4 APPROACH: SUCCESSIVE ACTORS FOR VALUE OPTIMIZATION (SAVO)

We aim to reformulate the actor-critic framework with the goal of finding actions that better maximize the critic's Q-function. As discussed in Section 3.2, actors trained with gradient ascent over the action-value landscape are susceptible to local optima. Our key innovation is to use an ensemble of successive actor-critic modules that collectively navigate the action-value landscape more proficiently than a single conventional actor. To distinguish from the additional critics, we refer to the original

---

**Algorithm 1** Successive Actors for Value Optimization (SAVO)

---

**Require:** State space $\mathcal{S}$, Action space $\mathcal{A}$, Primary critic $\mathcal{Q}$, Number of actor-critic pairs $K$
1: Initialize primary actor-critic pair $(\pi_0, Q_0) \leftarrow (\pi, \mathcal{Q})$
2: $\Pi(s)$ is the policy that selects $\hat{a_t^*} = \arg\max_{a \in A} \mathcal{Q}(s, a)$ where $A = \{\pi_0(s), \dots, \pi_{K-1}(s)\}$
3: Initialize actor-critic pair $(\pi_i, Q_i) \forall i = 1 \dots K - 1$
4: **for** timestep $t$ **do**
5:     Observe state $s_t$
6:     Initialize action list $A \leftarrow \emptyset$
7:     **for** $i = 0 \dots K - 1$ **do**
8:         Concatenate $A$ with state $s$, obtaining deep-set summary $Z$
9:         Condition $\pi_i$ and $Q_i$ with FiLM layer based on $Z$
10:        Select action $a_i \sim \pi_i(s, Z)$ ; Add OU Noise for exploration
11:        Append $a_i$ to $A$
12:     **end for**
13:     Select $\hat{a_t^*} = \arg\max_{a \in \{a_1, \dots, a_k\}} \mathcal{Q}(s_t, a)$
14:     Execute $\hat{a_t^*}$, observe reward $r_t$ and new state $s_{t+1}$
15:     **for** $i = 0 \dots K - 1$ **do**
16:        Update $Q_i$: $Q_i(s_t, A, a_i) = \max_{j=0 \dots i} (\mathcal{Q}(s_t, a_j))$
17:        Update actor $\pi_i$: $\nabla_{\phi_i} J(\phi_i) = \mathbb{E}_{s \sim p_\pi} \left[ \nabla_a Q_i(s_t, A, a) \big|_{a = \pi_i(s_t, A)} \nabla_{\phi_i} \pi_{\phi_i}(s_t, A) \right]$
18:     **end for**
19:     Update the primary critic with TD3: $\mathcal{Q}$ using: $\mathcal{Q}(s_t, a_t) = r_t + \gamma \mathcal{Q}(s_{t+1}, \Pi(s_{t+1}))$
20: **end for**

---

critic as the primary critic, $\mathcal{Q}$. Our proposed approach SAVO combines two solutions: explicitly using the primary critic to evaluate multiple action proposals and a novel training objective for successively making the Q-function landscape more tractable to optimize.

## 4.1 Primary Critic for Action Evaluation

The main objective of the actor is to find actions to maximize the $\mathcal{Q}$-function (Silver et al., 2014), $a^* = \arg\max_a \mathcal{Q}(s, a)$. An intuitive way to address the drawback of a single actor optimizing the critic is to utilize the critic to evaluate multiple action proposals. Therefore, if we can get $k$ action proposals, our bets are hedged and we are more likely to find an action closer to the maximum.

$$\hat{a^*} = \arg\max_{a \in \{a_1, a_2, \dots, a_k\}} \mathcal{Q}(s, a)$$

We can formulate this learning objective as finding a joint set of action proposals, such that one of them maximizes the primary critic. Thus, we can train a large joint actor-critic modeled with $\Pi_{\text{joint}}(\{a_1, a_2, \dots a_k\} | s)$ and $Q_{\text{joint}}(\{a_1, a_2, \dots a_k\}, s)$. Here, $\Pi_{\text{joint}}$ is trained just like TD3, and $Q_{\text{joint}}$ can be trained to imitate the Q-value of the best action:

$$\min_{\theta_{\text{joint}}} \mathbb{E}_{\{a_1, \dots, a_k\} \sim \Pi_{\text{joint}}, s \sim \text{Env}} \left[ \left( Q_{\text{joint}}(\{a_1, a_2, \dots, a_k\}, s; \theta_{\text{joint}}) - \max_{i=1}^{k} \mathcal{Q}(s, a_i) \right)^2 \right]$$

While this is a valid learned solution to our problem, it brings in new challenges because the action space for $\Pi_{\text{joint}}(\{a_1, a_2, \dots a_k\} | s)$ and $Q_{\text{joint}}(\{a_1, a_2, \dots a_k\}, s)$ is now large: $D * k$. A larger action space in the joint-actor-critic model present another optimization challenge, making this solution infeasible. The key observation here is that this large action space is actually separable into individual actions. This motivates our key reformulation into successive actor-critic modules.

## 4.2 Successive Actor-Critic Modules

We aim to break down the problem of jointly finding a good set of actions to explicitly evaluate the primary critic. Therefore, we propose to solve a sequence of MDPs with successively simpler optimization landscapes (Figure 1). We have a sequence of K actors $\pi_i$ and critics $Q_i$, where $i = 1 \dots K$ and the first actor and critic are identical to the primary critic. Our goal is to make the optimization landscapes of $Q_i$'s to be more tractable successively. We achieve this by:

1. Training each $Q_i$ with an objective to mimic the primary critic, but lower bounded by the previous best actions' $\mathcal{Q}$-value. This makes $Q_i$'s landscape with fewer local optima than prior critics in the sequence because the peaks below the lower bound baseline disappear. The actor $\pi_i$ can be trained with TD3 over $Q_i$, just like the primary actor-critic pair. However, the key difference is that $Q_i$ has fewer local optima than the primary critic $\mathcal{Q}$, which makes it easier for $\pi_i$ to find an effective action.

2. Conditioning each actor-critic module $\pi_i$ and $Q_i$ on the previously selected actions $\{a_0, \ldots, a_{i-1}\}$: (i) we choose a deep-set (Zaheer et al., 2017) to summarize the previous actions into an encoding (LSTM and Transformer are also valid choices) and (ii) use a FiLM conditioning layer (Perez et al., 2018) to condition the actor and critic networks with this action summary encoding (simply concatenating with state input is also a valid choice). More details and empirical study of these design choice are in Appendix D, D.

Thus, our final training objective is:

$$Q_i(s, \{a_0, \ldots, a_{i-1}\}, a_i) = \max_{j=0\ldots i} (\mathcal{Q}(s, a_j))$$

$$\nabla_{\phi_i} J(\phi_i) = \mathbb{E}_{s \sim p_\pi} \left[ \nabla_a Q_i(s, \{a_0, \ldots, a_{i-1}, a\}) \big|_{a=\pi_i(s, \{a_0, \ldots, a_{i-1}\})} \nabla_{\phi_i} \pi_{\phi_i}(s, \{a_0, \ldots, a_{i-1}\}) \right]$$

$$\mathcal{Q}(s, a) = r + \gamma \mathcal{Q}(s', a'), \quad a' = \arg\max \mathcal{Q}(s', \Pi(s')),$$

where $\Pi(s') = \arg\max_{a \in \{\pi_0(s'), \ldots, \pi_{K-1}(s')\}} \mathcal{Q}(s', a)$

Our approach (elaborated in Algorithm 1 has several benefits:

- Improved exploration with OU noise centered over an action with higher Q-value.
- Updates are done on more optimal target actions as opposed to a naive actor.
- The architecture reduces to the primary actor-critic TD3 algorithm for K = 1, which means we are lower-bounded by the baseline TD3 performance in our ability to maximize the critic.

## 5 ENVIRONMENTS

We evaluate SAVO on discrete and continuous action space environments with challenging optimization landscapes of the Q-value (Figure 5). More environment details are present in Appendix B.

### 5.1 DISCRETE ACTION SPACE

**Mining Task:** A modified grid world environment (Chevalier-Boisvert et al., 2018) where the red agent (upper left in (a) of Fig.5) navigates a 2D maze to read the green goal (bottom right), removing mines with pick-axes to reach a goal as quickly as possible. There are two kinds of actions, navigation and mine breaking, enabling the agent to reach the goal as soon as possible. The action space includes navigation and tool actions. Successful digging and reaching the goal yield rewards. The action representations introduce complexity, and the distance in the action representation space is unreliable.

**Simulated and Real-Data Recommender Systems:** RecSim (Ie et al., 2019) simulates user interactions in a recommender system with a large discrete action space. The agent must recommend the most relevant item from a set of 10,000 items based on user preference information. The environment assumes full observability (*RecSim* in Fig.2) with the state as the user preference vector and action representations as item characteristics, or more real-world setting (*RecSim-Data* in Fig.2) using the real movies from Movielens (Harper and Konstan, 2015) as recommendation items.

### 5.2 CONTINUOUS CONTROL

Suite of MuJoCo (Todorov et al., 2012) continuous control tasks interfaced through OpenAI Gym. The environments include Ant, HalfCheetah, Hopper, Humanoid, Walker2D, Inverted Pendulum, and Inverted Double Pendulum. The actor's ability to maximize the Q-function is tested against baselines, and discontinuities in the action space are introduced by inducing regions of validity outside which actions are not executed in the environment, called *Hard* environments as discussed in Sec.6.

## 6 EXPERIMENTS

We design experiments to answer the following questions in the context of RL with challenging innumerable action spaces: (1) How effective is our method in challenging action spaces? (2) Does SAVO effectively optimize the critic and environment reward? (3) How robust is SAVO to increasingly challenging value landscapes?

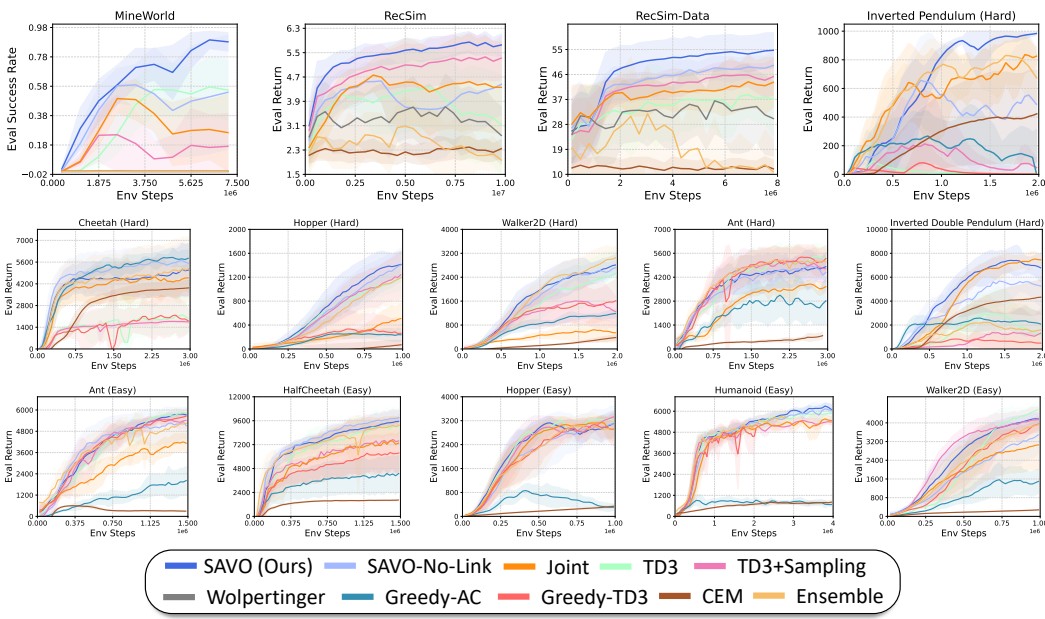

Figure 2: We evaluate SAVO against baselines on the challenging action space. The results are averaged over 5 random seeds, and the seed variance is shown with shading.

## 6.1 EFFECTIVENESS OF SAVO ON CHALLENGING ACTION SPACES

### 6.1.1 BASELINES AND ABLATIONS

**k-Nearest Neighbors in Action Space**: (1) *Wolpertinger* follows (Dulac-Arnold et al., 2015) to train a value-based actor-critic for the following two-phase framework. The k-nearest action representations to the actor's output are retrieved as candidates in which the critic finds the action with the maximum Q-value for the final decision-making. The critic also provides gradients to train the actor. (2) *TD3+Sampling* is a robust and stronger baseline that trains an extra critic only to train the actor (in contrast, Wolpertinger's critic makes decisions as well as provides the learning signal to the actor). In continuous action spaces, we sample $k$ actions in a Gaussian centered at the actor's output.

*CEM-Actors*: CEM-based actor following evolutionary approaches discussed in Section 2.4. *Greedy Actor-Critic*: Follows (Neumann et al., 2018) by training a high-entropy proposal policy. *Greedy TD3*: We implement a version of Greedy-AC with TD3 exploration and update improvements. *Ensemble Actor-Critic*: An ensemble (Osband et al., 2016) of actors are trained with the same primary critic, without SAVO's successive critics to prune the optimization landscape.

**SAVO Ablations**: *SAVO (Ours)* learns a cascaded actor-critic architecture for retrieving $k$ actions. (1) *SAVO-Joint (or Joint)* is $\Pi_{\text{joint}}$ introduced in Section 4.1 that trains an actor with a large joint action space of $k \times D$. The $k$ action samples are obtained by splitting the joint action into $D$ dimensions. However, without splitting into successive actors, the action space is large and, as a consequence, is expected to face difficulty in learning. (2) *SAVO-no-link* removes the list encoder in *SAVO*, such that each actor-critic pair acts as an independent agent. This results in a multi-agent network without any communication. Thus, this does not inform the subsequent actor-critic modules of previously selected actions, but the modules are still updated with a pruning objective. (3) *SAVO w/h List-Length=1 (i.e., TD3)* only incorporates the primary actor-critic in *SAVO*, which reduces to the base algorithm, TD3.

### 6.1.2 RESULTS

- **Mine World**: Figure 2 shows that *SAVO* outperforms all the baselines. *SAVO-ablations* are able to achieve reasonable success rates as well. It is a challenging action space as two categorically different kinds of discrete actions are available to the agent - navigation and mining actions. *SAVO*'s efficacy demonstrates that the successive actor-critic modules are able to navigate over local optima.

- **RecSim**: We observe that *SAVO* was able to outperform all the baselines, while *TD3+Sampling* was able to achieve reasonably well performance but did not achieve the same optimality. *SAVO w/o cascading* converges suboptimally due to independent action retrieval. *Joint* is slow to learn due to

the large actor-critic agent that needs to be trained. Interestingly, we observe that TD3+Sampling outperforms Wolpertinger, which shows the importance of learning an extra critic just for training the retrieval actor, unlike what was originally proposed in Dulac-Arnold et al. (2015).

- **Continuous Control**: In the *Hard* tasks (*Hopper-Hard* and *Walker2D-Hard*), we observe that *SAVO* outperforms all the baselines with a good sample efficiency in learning. *TD3+Sampling* and *SAVO-ablations* shows sign of learning but they did not reach the same optimality as our method in the limited training time-budget because of the large discontinuity in the action space. *SAVO* has actor-critic agents that can explore different parts of the action space and propose multiple candidates to the selection Q-network. This is also the reason why the ablation of SAVO without cascading still does reasonably well because of multiple candidates in exploration. SAVO still outperforms due to its linkage of actors. In *Easy*-continuous control tasks, the findings revealed that baseline models consistently performed better than more challenging tasks (*Hard*). This suggests that, in the intricate continuous control tasks, our approach demonstrates superior adaptability and effectiveness and the trend is consistent when the task become easier.

## 6.2 OPTIMIZATION ANALYSIS: ARE Q-VALUE LANDSCAPES EFFECTIVELY PRUNED?

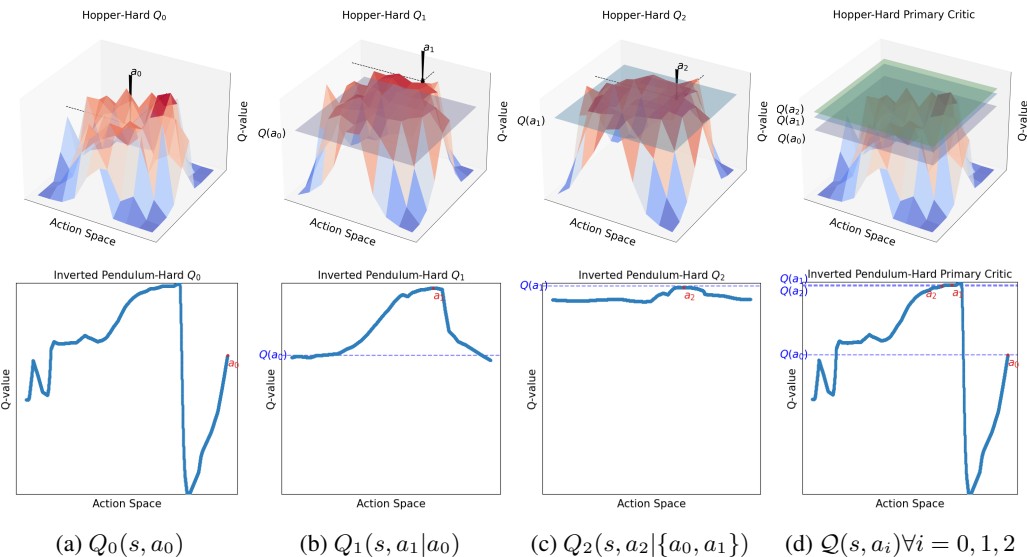

(a) $Q_0(s, a_0)$  (b) $Q_1(s, a_1|a_0)$  (c) $Q_2(s, a_2|\{a_0, a_1\})$  (d) $\mathcal{Q}(s, a_i)\forall i = 0, 1, 2$

Figure 3: Analysis of SAVO's Q-value landscapes.

Figure 3 illustrates the efficacy of SAVO in pruning the Q-optimization landscape, resulting in improved global optima for *Hopper-Hard* (*Top row*) and *Inverted Pendulum-Hard* (Bottom row; (a, b, c, d)). The sequence of plots ( *(a) $\sim$ (c)* ) demonstrates the stepwise removal of local optima by $Q_0(s, a_0)$, $Q_1(s, a_1|a_0)$, and $Q_2(s, a_2|\{a_0, a_1\})$. In plot *(d)*, successive actors successfully identify $a_0, a_1, a_2$ with progressively better $\mathcal{Q}$-values, showcasing their ability to globally optimize the primary critic more effectively than a single actor. Additionally, the surfaces represented by $Q_0$, $Q_1$, and $Q_2$ exhibit a trend toward increased smoothness. In *Hopper-Hard* environment plots, a growing prevalence of warm colors indicates higher overall Q-values, and in *Inverted Pendulum-Hard* environment plots by a reduced number of peaks, further confirms this upward trend. Additional results can be found in Fig 10 and Fig 11.

### 6.2.1 AGGREGATED RESULTS: PERFORMANCE PROFILES

Agarwal et al. (2021) proposed a robust means to rigorously validate the efficacy of our approach. Through the incorporation of the suggested performance profile, we have conducted a more thorough comparison of our approach against baselines, resulting in a comprehensive understanding of the inherent statistical uncertainty in our results. In Figure 4c, the x-axis illustrates normalized scores across all tasks, employing *min-max scaling* to normalize scores based on the initial performance of untrained agents aggregated across random seeds (i.e., *Min*) and the final performance presented in Figure 2 (i.e., *Max*).

Figure 4c reveals the consistent high performance of our method across various random seeds, with its curve consistently ranking at the top of the x-axis changes, while baseline curves exhibit earlier declines compared to our approach. This visual evidence substantiates the robustness and reliability of our method across different experimental conditions.

## 6.3 QUALITATIVE ANALYSIS

**[Analysis 1] If action-value has improved over the preceding actors?**: Figure 4a provides a comparative analysis between our approach, which focuses on learning to identify multiple advantageous actions, and the baseline method that repetitively samples actions in close proximity to a single selected action. Our observations reveal that our method exhibits superior enhancements in action value when contrasted with the baseline. Consequently, this demonstrates the consistent and effective capability of our method to maximize the Q-function in comparison to a single actor-critic model.

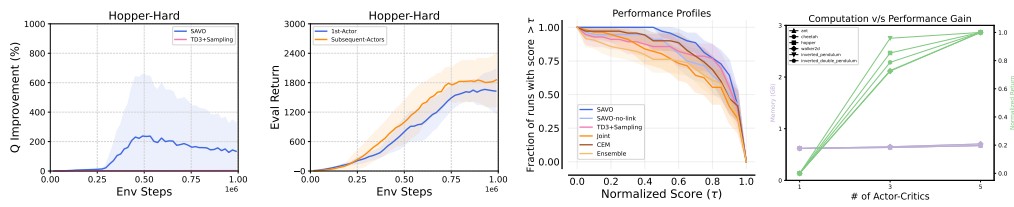

| (a) Q-Improvement % | (b) Return Improvement | (c) Performance Profiles | (d) Comp. v/s Perf. |

Figure 4: Qualitative analyses of SAVO. Results are aggregated over 5 random seeds.

**[Analysis 2] Does Q-value maximization lead to better environment return?**: Figure 4b shows the difference in the following two types of evaluation episode-return for our method; (i) Our method only uses the first actor in decision-making, (ii) Our method uses the full set of actors in decision-making. This comparison shows that our method is able to leverage the subsequent actors in decision-making to achieve better optimality. In Figure 4 (Right), it becomes apparent that the initial performance of actors in our method is similar. However, as the training progresses, we observe a widening disparity between the episode return based on (i) and (ii). This phenomenon underscores the fact that the actors within our method progressively evolve to make significant contributions to the decision-making process, leading to improved performance outcomes.

## 6.4 ANALYSIS: VARYING COMPLEXITY OF ACTION SPACES

We test the robustness of our method to more challenging Q-value landscapes in Figure 7 in Appendix.D.1. In RecSim, we vary the action space size, from $100K$ to $500K$. The results show that SAVO outperforms the baselines, maintaining its robust performance even as the action complexity increases. In contrast, the baselines experienced performance deterioration as action sizes grew larger.

## 6.5 ANALYSIS: PERFORMANCE V/S COMPUTATION GAINS

In this experiment (Fig.4d), the impact of varying the number of actor-critic pairs (X-axis) on the performance (converged episode return (Right Y-axis)) and the memory cost (Left Y-axis) across different Mujoco (*Hard*) environments with 1, 3, and 5 actors-critics is investigated. We verified our hypothesis that the performance gain would surpass the memory gain across the different numbers of actor-critics. We also found that the potential optimality cap, irrespective of the number expansion, implying that further increasing in actor-critics would not yield proportional benefits.

## 7 CONCLUSION

We identify and address the problem of ineffective optimization of the critic in conventional single actor-critic algorithms. We present Successive Actors for Value Optimization (SAVO), an approach that effectively maximizes the critic, leading to improved sample efficiency and final performance in several large-discrete action space and continuous action space RL environments.

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

APPENDIX

# A  REPRODUCIBILITY STATEMENT

With the aim of promising the reproducibility of our results, we have attached our code in the supplementary materials, which contain all environments and all baseline methods we report in the paper. The specific commands to reproduce all baselines across all environments are available in README. We have also included all relevant hyperparameters and additional details on how we tuned each baseline method in Appendix Section 1.

# B  ENVIRONMENT DETAILS

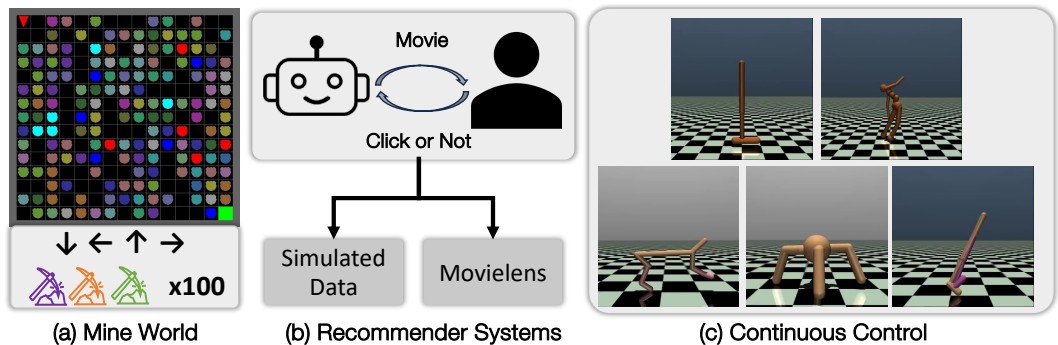

(a) Mine World  (b) Recommender Systems  (c) Continuous Control

Figure 5: This figure provides the visual description of the environment setup.

## B.1  MININGENV

The grid world environment, introduced in Sec. 5.1, requires an agent to reach a goal by navigating a 2D maze as soon as possible while breaking the mines blocking the way.

**State**: The state space is an 8+K dimensional vector, where K equals to *mine-category-size*. This vector consists of 4 independent pieces of information: Agent Position, Agent Direction, Surrounding Path, and Front Cell Type.

1. Agent Position: Agent Position occupies two dimensional of the vector. The first dimension represents the x-axis value, and the second one represents the y.

2. Agent Direction: It only takes one channel with value [0, 1, 2, 3]. Each number represents one direction, and they are 0-right, 1-down, 2-left, and 3-up.

3. Surrounding Path: This information takes four channels. Each represents whether the cell in that direction is an empty cell or a goal.

4. Front Cell Type: This information is in one-hot form and occupies the last $K + 1$-dimensional vector, which provides the information of which kind of mine is in front of the agent. If the front cell is an empty cell or the goal, the $(K + 1)^{th}$ channel will be one, and others remain to be zero

Ultimately, we will normalize each dimension to [0, 1] with each dimension's minimum/maximum value. Each time we reset the environment, the layout of the whole grid world will be changed, except for the agent start position and the goal position.

**Termination**: An episode is terminated in success when the agent reaches the goal or after a total of 100 timesteps.

**Actions**: The base action set combines two kinds of actions: navigation actions and pick-axe(tool) actions. The navigation action set is a fixed set, which contains four independent actions: going up, down, left, and right, corresponding with the direction of the agent. They will change the agent's direction first and then try to make the agent take one step forward. Note that, different from the

empty cell, the agent cannot step onto the mine, which means that if the agent is trying to take a step towards a mine or the border of the world, then the agent will stay in the same location while the direction will still be changed. Otherwise, the agent can step onto that cell. An agent will succeed if it reaches the goal position. The size of the pick-axe action set is equal to 50. Each tool has a one-to-one mapping, which means they can and only can be successfully applied to one kind of mine, and either transform that kind of mine into another type of mine or directly break it.

**Reward**: The agent receives a large goal reward for reaching the goal. The goal reward is discounted based on the number of action steps taken to reach that location, thus rewarding shorter paths. To further encourage the agent to reach the goal, a small exploration reward is added whenever the agent gets closer to the goal, and a negative equal penalty is added whenever the agent gets further to the goal. And also, when the agent successfully applies a tool, it will gain a small reward. When the agent successfully breaks a mine, it will also gain a small bonus.

$$
\begin{aligned}
R(s,a) \ = \ & \mathbb{1}_{Goal} \cdot R_{\text{Goal}} \left( 1 - \lambda_{\text{Goal}} \frac{N_{\text{current steps}}}{N_{\text{max steps}}} \right) + \\
& R_{\text{Step}} \left( D_{\text{distance before}} - D_{\text{distance after}} \right) + \\
& \mathbb{1}_{correct\ tool\ applied} \cdot R_{\text{Tool}} + \\
& \mathbb{1}_{successfully\ break\ mine} \cdot R_{\text{Bonus}}
\end{aligned}
\tag{3}
$$

where $R_{\text{Goal}} = 10$, $R_{\text{Step}} = 0.1$, $R_{\text{Tool}} = 0.1$, $R_{\text{Bonus}} = 0.1$, $\lambda_{\text{Goal}} = 0.9$, $N_{\text{max steps}} = 100$

**Action Representations**: The action representations are 4-dimensional vectors manually defined using a mix of number ids, and each dim is scaled to [0, 1]. as shown in Graph 6. Dimensions 1 identifies the category of skills (navigation, pick-axe), 2 distinguishes movement skills (right, down, left, up), 3 denotes the mine on which this tool can be successfully applied, and 4 shows the result of applying this tool. We will normalize the action embedding space to [0, 1] for each dimension.

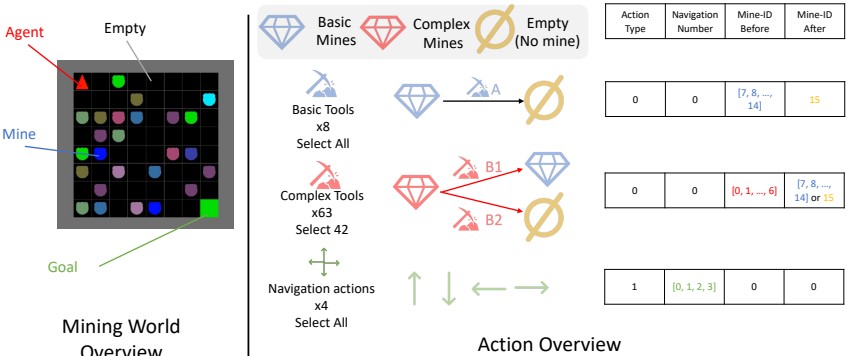

Figure 6: Mining Env Setting Description

## B.2 RECSIM

The simulated RecSys environment requires an agent to select an item that match the user's interest out of a large item-set. We simulate users with a dynamically changing preference upon clicks. Thus, the agent's task is to infer this preference from user clicks and recommend the most relevant item to maximize a total number of clicks.

**State**: The user interest embedding ($e_u \in \mathbb{R}^n$ where $n$ denotes the number of categories of items) represents the user interest in categories that transitions over time as the user consumes different items upon click. So, when the user clicks an item with the corresponding item embedding($e_i \in \mathbb{R}^n$; the same $n$ as the one for the user embedding) then the user interest embedding($e_u$) will be updated

as follows;

$$\Delta(e_u) = (-y|e_u| + y) \cdot (1 - e_u), \text{ for } y \in [0, 1]$$
$$e_i \leftarrow e_u + \Delta(e_u) \text{ with probability}[e_u^T e_i + 1]/2$$
$$e_u \leftarrow e_u - \Delta(e_u) \text{ with probability}[1 - e_u^T e_i]/2$$

This essentially pulls the user's preference towards the item that was clicked.

**Action**: The action set contains many recommendable items. So, the agent has to find the most relevant item to a user given the item-set. See below regarding how these items are represented.

**Reward**: The base reward is a simulated user feedback (e.g., clicks). The user model (Ie et al., 2019) stochastically skips or clicks the recommended item based on the present user interest embedding ($e_u$). Concretely, the user model computes the following score on the recommended item;

$$\text{score}_{item} = \langle e_u, e_i \rangle$$
$$p_{item} = \frac{e^{score_{item}}}{e^{s_{item}} + e^{score_{skip}}}$$
$$p_{skip} = \frac{e^{score_{skip}}}{e^{s_{item}} + e^{score_{skip}}}$$

where, $e_u, e_i \in \mathbb{R}^n$ are the user and item embedding, respectively, $\langle \cdot, \cdot \rangle$ is the dot product notation and $score_{skip}$ is a empirically decided hyper-parameter. So, given the score $score_{item}$ of an item, the user model computes the click likelihood through a softmax function over the recommended item and a predefined skip score. Finally, the user model stochastically selects either click(reward=1) or skip(reward=0) based on the categorical distribution on $[p_{item}, p_{skip}]$.

**Action Representations**: Following Jain et al. (2021), we implement continuous item representations sampled from a Gaussian Mixture Model (GMM) with centers around each item category. In this work, we did not use the sub-category in the category system.

### B.3 CONTINUOUS CONTROL

The MuJoCo (Todorov et al., 2012) benchmarking tasks are a set of standard reinforcement learning environments provided by the MuJoCo physics engine. elow is a brief description of some of the commonly used MuJoCo benchmarking tasks:

**Hopper**: In the Hopper task, you control a one-legged robot that must learn to hop forward while maintaining balance. The agent needs to find an optimal hopping strategy to maximize forward progress.

**Walker2d**: This task features a two-legged robot that must learn to walk forward. Similar to the Hopper, the agent must maintain balance while moving efficiently.

**HalfCheetah**: The HalfCheetah task involves a four-legged cheetah-like robot. The objective is for the robot to learn a coordinated gait that allows it to move forward as rapidly as possible.

**Ant**: In the Ant task, you control a four-legged ant-like robot. The challenge is for the robot to learn to walk and navigate efficiently through its environment.

**Humanoid**: The Humanoid task features a bipedal humanoid robot with many degrees of freedom. The agent must learn to control the complex humanoid body to walk, run, and perform other movements.

These MuJoCo benchmarking tasks vary in complexity and offer a diverse set of challenges for reinforcement learning agents.

## C    NETWORK ARCHITECTURES

### C.1    SUCCESSIVE ACTORS

The whole actor has a successive format and each successive actor will receive two pieces of information: the state observation and the action list generated by previous successive actors. Given the concatenation of the input components above, a 4-layer MLP with ReLU will process this information and generate one action for one single successive actor. And this action will be concatenated with the previous action list. After being transformed by an optional action-list-encoder, together with the state information, they become the input of next successive actor's input. In the end, the action list will be processed with 1-NN to find the nearest discrete action. After this, this action list will be delivered to the selection Q-network.

### C.2    SUCCESSIVE CRITICS

The critic has a one-to-one mapping relationship with the actor. The whole critic consists of a list of successive critics and each successive critic will receive three pieces of information: the state observation, the action list generated by previous successive actors, and the action provided by the corresponding successive actor. Given the concatenation of the input components above, a 2-layer MLP with ReLU will process this information and generate the action's value for one single successive actor. This value will be used to update itself and the actor with TD-error.

### C.3    LIST SUMMARIZERS

In order to extract meaningful information from the list of candidate actions, following Jain et al. (2021) we employed the sequential models and the list-summarizer as follows;

**Bi-LSTM**: The raw action representations of candidate actions are passed on to the 2-layer MLP followed by ReLU. Then, the output of the MLP is processed by a 2-layer bidirectional LSTM (Huang et al., 2015). Another 2-layer MLP follows this to create the action set summary to be used in the following successive actor.

**DeepSet**: The raw action representations of candidate actions are passed on to the 2-layer MLP followed by ReLU. Then, the output of the MLP is aggregated by the mean pooling over all the candidate actions to compress the information. Finally, the 2-layer MLP with ReLU provides the resultant action summary to the following successive actor.

**Transformer**: Similar to the Bi-LSTM variant of the summarizer, we employed the 2-layer MLP with ReLU before inputting the candidate actions into a self-attention and feed-forward network to summarize the information. Afterward the summarization will be part of the input of the following successive actor.

### C.4    FEATURE-WISE LINEAR MODULATION (FiLM)

Feature-wise Linear Modulation (Perez et al., 2018), is a technique commonly applied in neural networks for tasks like image recognition. It enhances adaptability by dynamically adjusting intermediate feature representations. Using learned parameters from one layer, FiLM linearly modulates features in another layer, allowing the network to selectively emphasize or de-emphasize aspects of the input data. This flexibility is beneficial for capturing complex and context-specific relationships, improving the model's performance in various tasks.

### C.5    SELECTION Q-NETWORK

The selection Q-network sequentially evaluates the Q-value of the retrieved candidate actions by the cascading actors. Thus, it receives a concatenated information of state and an action embedding for each candidate action. Then, it selects the action with the largest Q-value amongst candidate actions to act on the environment.

# D  MORE EXPERIMENTAL RESULTS

## D.1  FIGURES OF ANALYSIS (VARYING COMPLEXITY OF ACTION SPACES)

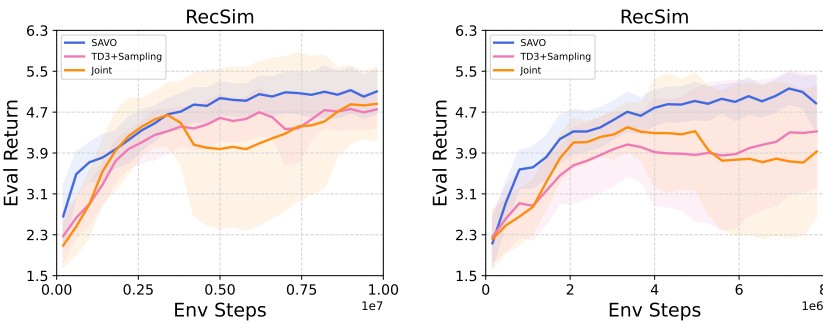

Figure 7: Increasing RecSim action set size: (Left) $100K$ items, (Right) $500K$ items (6 seeds).

## D.2  DESIGN CHOICES: ACTION SUMMARIZERS

In the exploration of action summarizer design choices, three key architectures were considered: Deepset, LSTM, and Transformer models, each represented by SAVO, SAVO-lstm, and SAVO-transformer in Fig.8, respectively. In the discrete tasks, the comparison revealed a preference for the deepset architecture over LSTM and Transformer. In the continuous domain, however, the results were rather varied, indicating that the effectiveness of the action summarizer depends on the specific use case. The nuanced differences among these architectures contribute to the complexity of the task, and further research is needed to determine the optimal design for action summarization in both discrete and continuous contexts.

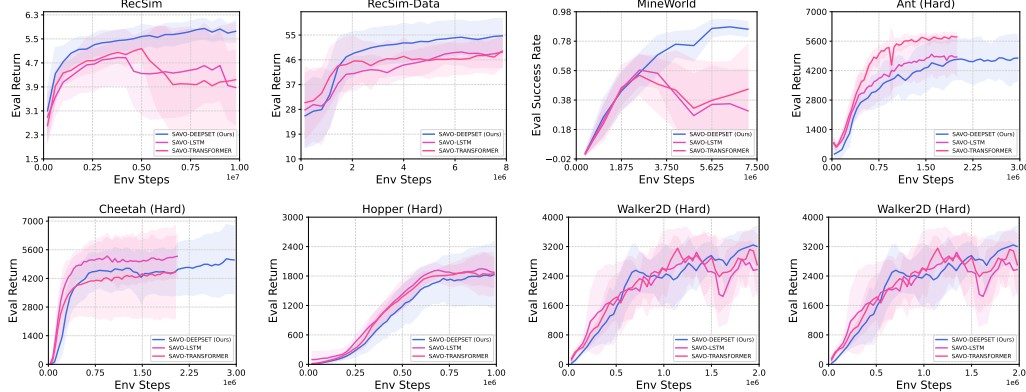

Figure 8: Comparison of action summarizers: the results are averaged over 5 random seeds, and the seed variance is shown with shading.

## D.3  CONDITIONING ON PREVIOUS ACTIONS: FiLM V/S MLP

In the examination of conditioning on previous actions, two distinct approaches, Feature-wise Linear Modulation (FiLM) and Multi-Layer Perceptron (MLP), represented by FiLM and non-FiLM variants in Fig.9, were scrutinized for their efficacy. In the discrete tasks, the results unveiled a notable preference for FiLM over non-FiLM implementations, highlighting its effectiveness in leveraging information from prior actions for improved conditioning. However, in the continuous domains, the comparison between FiLM and MLP yielded varied outcomes, suggesting that the choice between these approaches is intricately tied to the specific task context. The nuanced performance differences observed underscore the need for continued research to ascertain the optimal approach for conditioning on previous actions and to enhance model adaptability across diverse applications.

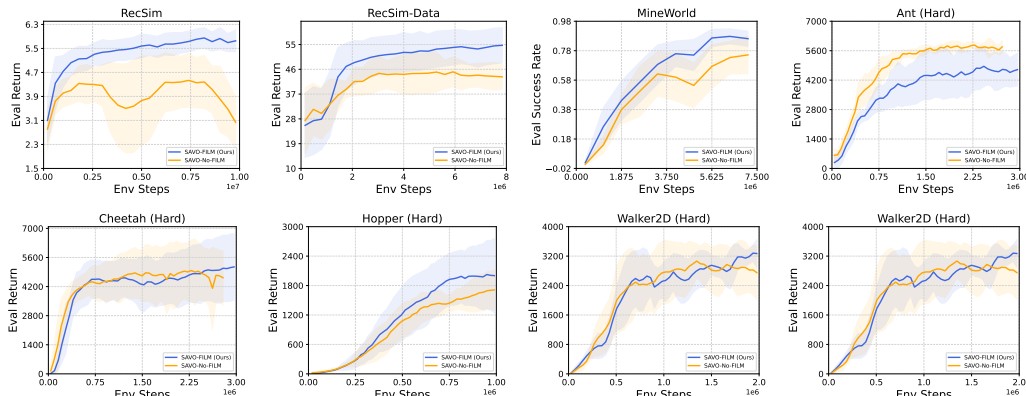

Figure 9: Comparison of how to condition on previous actions: the results are averaged over 5 random seeds, and the seed variance is shown with shading.

### D.4 Q-VALUE LANDSCAPE

We conducted a Q-space analysis across Mujoco environments to show that successive critics reduce local optima, aiding actors in optimizing actions. The outcomes are depicted in Figures 10 and 11.

Figure 10 illustrates a representative Q landscape from the easy environments, which are uniformly smooth. This uniformity in the primary Q space simplifies the identification of optimal actions.

Figure 11 shows that the primary Q landscape (leftmost and rightmost) in challenging environments is clearly uneven with several local optima. However, the Q landscapes learned by successive critics $Q_i$ demonstrate a gradual transition toward smoothness by pruning out the locally optimal peaks below the previously selected actions' Q-values. This aids the actors in identifying improved actions that are better global optima over the primary critic. Finally, when visualized together on the primary critic (rightmost figure) the subsequent actions yield more enhanced Q-values than $a_0$, which would have been the action selected by a single actor. This translates into better evaluation performance as shown in Figure 4b and better sample efficiency as shown in Figure 2.

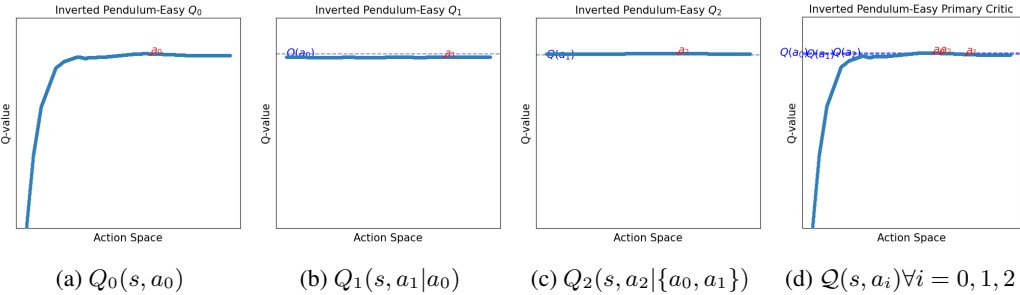

(a) $Q_0(s, a_0)$     (b) $Q_1(s, a_1|a_0)$     (c) $Q_2(s, a_2|\{a_0, a_1\})$     (d) $\mathcal{Q}(s, a_i)\forall i = 0, 1, 2$

Figure 10: Successive Q landscape and primary Q landscape of Inverted Pendulum-Easy.

## E EXPERIMENT DETAILS

### E.1 IMPLEMENTATION DETAILS

We used PyTorch (Paszke et al., 2019) for our implementation, and the experiments were primarily conducted on workstations with either NVIDIA GeForce RTX 2080 Ti, P40, or V32 GPUs on. Each experiment seed takes about 4-6 hours for Mine World, 12-72 hours for Mujoco, and 6-72 hours for RecSim, to converge. We use the Weights & Biases tool (Biewald, 2020) for plotting and logging experiments. All the environments were developed using the OpenAI Gym Wrapper (Brockman et al., 2016). We use the Adam optimizer (Kingma and Ba, 2014) throughout.

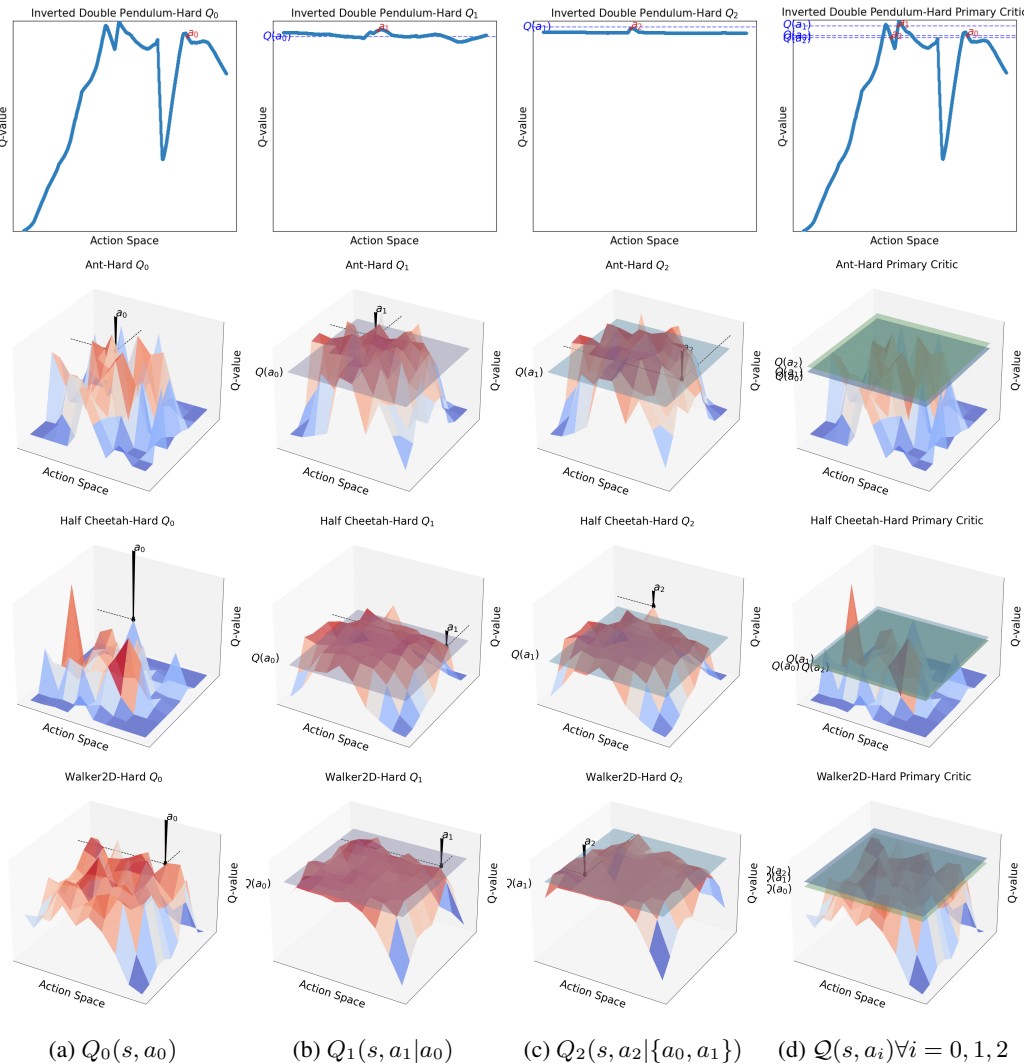

(a) $Q_0(s, a_0)$  (b) $Q_1(s, a_1|a_0)$  (c) $Q_2(s, a_2|\{a_0, a_1\})$  (d) $\mathcal{Q}(s, a_i)\forall i = 0, 1, 2$

Figure 11: Successive Q landscape and primary Q landscape across different Hard Environments.

### E.2 HYPERPARAMETERS

The environment-specific and RL algorithm hyperparameters are described in Table 1.

### E.3 COMMON HYPERPARAMETER TUNING

To ensure fairness across all baselines and our methods, We searched over hyper-parameters that are common across baselines;

- **Learning rate of Actor and Critic**: *(Actor)* We searched over $\{0.01, 0.001, 0.0001\}$ and found that $0.0001$ to be the most stable for the actor's learning in the discrete RL tasks and $0.001$ for Mujoco environments. *(Critic)* We searched over $\{0.01, 0.001, 0.0001\}$ and found that $0.001$ to be the most stable for the critic's learning across all tasks.

- **Network Size of Actor and Critic**: *(Critic)* In order for the fair comparison, we employed the same network size for the DQN components (ie., DQN, the selection network in the retrieval-selection framework, and the cascading/joint/extra critics in SAVO and the baselines). We individually performed the architecture search on each task and found a specific network size performing the best in the task. *(Actor)* Similarly to critic, we employed the same network size for the actor components in the baseline and the cascading actors in SAVO. And, likewise, we performed the

individual architecture search on each task and found a specific network size performing the best in the task.

| Hyperparameter | Mine World | MUJOCO | RecSim |
|---|---|---|---|
| **Environment** | | | |
| total timesteps | 10M | 10M | 10M |
| number of epochs | 5K | 8K | 10K |
| parallel processes | 5 | 10 | 16 |
| max episode steps | 100 | 500 | 20 |
| number of actions | 54 | N/A | 10000 |
| true action dimension size | 4 | 5 | 30 |
| extra action dimension size | 5 | N/A | 15 |
| **RL Training** | | | |
| actor lr | 0.001 | 0.001 | 0.0001 |
| retrieve critic lr | 0.001 | 0.001 | 0.001 |
| critic lr | 0.001 | 0.001 | 0.001 |
| list encoder lr | 0.0005 | 0.0005 | 0.0005 |
| actor hidden dimension | 128_64_64_32 | 64_64 | 64_32_32_16 |
| critic hidden dimension | 128_128 | 400_300 | 64_32 |
| batch size | 256 | 256 | 256 |
| buffer size | 500K | 500K | 1M |
| critic gamma | 0.99 | 0.99 | 0.99 |
| retrieve critic gamma | 0.99 | 0.99 | 0.9967 |
| explore with both actor and critic | TRUE | TRUE | TRUE |
| epsilon critic start value | 1 | 1 | 1 |
| epsilon critic end value | 0.01 | 0.01 | 0.1 |
| epsilon critic decay steps | 5M | 500K | 4M |
| epsilon actor start value | 1 | 1 | 1 |
| epsilon actor end value | 0.01 | 0.001 | 0.1 |
| epsilon actor decay steps | 5M | 500K | 4M |
| actor eval epsilon | 0 | 0 | 0 |
| critic eval epsilon | 0 | 0 | 0 |
| number of update times per epoch | 20 | 50 | 20 |
| **List Encoder Architecture** | | | |
| list length | 4 | 3 | 3 |
| actor weight initialization | none | none | add |
| critic weight initialization | none | none | add |
| list encoder | deepset | non-shared-deepset | deepset |

Table 1: Environment/Policy-specific Hyperparameters

