# OpenReview forum: "Rethinking Actor-Critic: Successive Actors for Critic Maximization"
_ICLR.cc/2024/Conference — Submitted to ICLR 2024_

### Official Review · Reviewer_ojfo · 2023-10-24

**Soundness:** 3 good
**Presentation:** 3 good
**Contribution:** 3 good
**Rating:** 5
**Confidence:** 4

**Summary:**

The paper presents a novel reformulation of the actor by employing a sequence of sub-actors, which solves the problem of non-convex and high-dimensional, and non-stationary during action-value optimization landscape training. The logical assumption stated by the authors is an ensemble of successive actor-critic modules can collectively navigate the action-value landscape more proficiently than a single, monolithic actor. The authors demonstrate improvement over continuous and large discrete action space reinforcement learning tasks.

**Strengths:**

The paper is well written and well structured. The idea of successive actor modules for "pruning" all actions with Q-values lower than the baseline is interesting and (to the best of my knowledge) novel. The experimental setup (especially on large discrete action space RL tasks and more discontinuous variants of continuous RL tasks) appears rigorous.

**Weaknesses:**

1.The paper's central assumption feels reasonable, and the experiment seems to confirm it. But there is no theoretical proof.
2.The key parts of the successive actor-critic modules adopt both ‘deep-set’ and ‘FiLM’ methods, but lack a description of potential advantages and an explanation of alternative methods.
3.From the structure of the proposed method, it can fluidly integrated into other widely adopted RL algorithms. In the experiment, TD3 was selected as the baseline. Is it possible to add other RL methods to the ablation study to illustrate the applicability of the method.
4.FIG. 1 is a diagram illustrating the core ideas of the paper. Can 'tractable' be explained from the perspective of real data in the experimental part? FIG. 4 seems intended to explain, but is insufficient.

**Questions:**

1.Algorithm 1, line 17 has a prominent '{' symbol. What is the difference between $(s_t|A)$ and $(s_t,A)$ in formulas $a=\pi_{i}(s_t|A)$ and $\pi_{\phi_i}(s_t,A)$.
2.Legend should be added to FIG. 7, although it is related to FIG. 3.
3.Although there is not enough time to go through the source code carefully, it is recommended that the method abbreviation be consistent in the code(called FLAIR) and the paper(called SAVO).

---

> ### Author Response · Authors · 2023-11-23
> **Thank you for your review! Here are our responses.**
>
> We thank you for your constructive feedback. We appreciate your positive comments on the approach and experimental performance. We address your concerns below:
>
> ### FiLM, DeepSet Design Choices
> - In Appendix C.3 and C.4, we added the descriptions of Deepset along with other design choices of list-summarizers as well as Feature-wise Linear Modulation (FiLM). In short, Deepset effectively aggregates the information in the list input while FiLM helps to condition on preceding list-input. Those modules contribute to better training the list of actor-critics in SAVO.
> - We also added ablations in Appendix.D.2 & D.3 to compare (i) FiLM v/s no-FiLM and (ii) the choice of DeepSet v/s LSTM v/s Transformers to aggregate the information from previous selected actions. The results validated our choice of DeepSet as it is more scalable to an increasing number of actor-critic pairs. Also, we acknowledge that this design choice depends on the environment and the use-case as some tasks in continuous domains show that LSTM / Transformers outperform Deepset.
>
> ### Visual insights on central claim
> `[New Figure.3]` The optimization landscape of Q-value concretely represents the landscape where the N-D action space represents the independent variable and the 1-D Q-value represents the dependent variable. For visualization, we map the action space to 2-D when N > 1. Fig.3 demonstrates that the Q-function becomes more easy to globally optmize with every iteration of our successive actor-critics. This is because the locally optimal peaks below past actions’ Q-values are pruned away from the optimization landscape. As we can observe:
> The number of local optima are reduced in $Q_0 \to Q_1 \to Q_2$, as most action values are shifted up in the Q-value space. This enables successive actors $\pi_0 \to \pi_1 \to \pi_2$ to focus on more optimal regions of the true Q-value space.
> The actions $a_1$ and $a_2$ found by $\pi_1$ and $\pi_2$ are indeed more optimal than $\pi_0$ whose actions were only locally optimal peaks, but globally suboptimal
>
> ### Clarifications of Baseline Selection
> We choose TD3 because of its unified applicability to both action-representation-based discrete action spaces and continuous action space. Particularly, the Wolpertinger algorithm [Dulac et al 2015] is based on the DDPG algorithm and is the most commonly used base algorithm for large discrete action spaces. While we agree with the reviewer’s suggestion that combining SAVO with SAC would be good to show wider applicability of our method, we think this extension might offer more flexibility in the design choices. Thus, for this project, we decided to stick to the simple architecture of TD3 as described above.
>
> ### Notational fixes
> We have fixed all the suggested typos, missing legends, and source code naming in our revision. Thank you for your detailed suggestions.
>
> Please do let us know if you have any leftover concerns or questions.

---

### Official Review · Reviewer_oxP1 · 2023-10-28

**Soundness:** 2 fair
**Presentation:** 3 good
**Contribution:** 2 fair
**Rating:** 5
**Confidence:** 3

**Summary:**

This paper studied an interesting problem that the training of the policy network often cannot effectively optimize the learned value function. This could lead to sub-optimal learning performance and ineffective exploration. To address this issue, this paper proposed a new ensemble technique that utilizes a sequence of separately trained actor-critic pairs to gradually refine/restrict the action space being considered. The newly proposed algorithm has been experimented on multiple benchmark problems with both discrete and continuous action spaces. The performance results showed that the new algorithm can achieve better overall performance across all the benchmark problems.

**Strengths:**

It is a well-known problem in the literature that the trained policy network in an actor-critic algorithm may often fail to optimize the learned value function. This inconsistency could potentially weaken the performance of the learning process. This paper developed an interesting new algorithm to address this issue. The effectiveness of the new algorithm is also evidenced by promising experiment results.

**Weaknesses:**

While the idea of using a series of successive actor-critic pairs to gradually refine/restrict the action space is interesting, however, this also means that the action selection decision from the policy network may be highly sensitive to the minor nuances of the learned critic. This often introduces bias to the learning process, resulting in degraded learning stability and restricted exploration. Hence, the downside of using multiple successive actor-critic pairs should be extensively examined in this paper. It is important to know why successively restricting the action space based on the trained critic will not affect the learning stability with a solid theoretical foundation. It is also important to know why this actually helps to improve the effectiveness of exploration, rather than restricting exploration, as claimed by the authors.

Since the newly proposed algorithm uses an ensemble of actor-critic pairs, it is related to ensemble actor-critic algorithms. Hence, in section 2, it seems necessary for the authors to review existing ensemble actor-critic methods and clearly highlight the key novelty of the new algorithm, compared to existing ensemble algorithms. Furthermore, the experiment study should include more state-of-the-art ensemble baselines, in order to clearly show the advantages of the new algorithm over existing ensemble algorithms.

Some parts of the new algorithm design seem to lack technical clarity. In particular, it is not clear to me how deep-set is used to produce Z as a concatenation of previously selected actions and state. It is also unclear how FiLM is used to enable a policy to choose its actions that are conditional on Z. Meanwhile, The motivations and rationales of using deep-set and FiLM should be clearly explained and strongly justified.

The authors stated that their new ensemble technique can be applied to many different actor-critic algorithms. Given this statement, it is not clear why they focus primarily on applying their new technique to TD3 alone. To demonstrate the wide applicability of the new technique, the authors should study its possible application to other algorithms, such as SAC.

I don't quite understand some mathematical formulas presented in this paper. For example, I don't know how to find the optimal action a' based on the primary critic, as part of the final training objective, which is further conditional on $\Pi$. The formula for policy gradient in the final training objective also misses some brackets such as }. Meanwhile, it remains questionable why the policy gradient formula is valid, i.e. what kind of gradient is being calculated and why the gradient allows the trained policy network to maximize the expected return. I think more detailed and thorough theoretical analysis is necessary to justify the validity and effectiveness of the proposed training objective.

Some statements in this paper are not easy to understand. For example, what does it mean by "navigate the action-value landscape more proficiently" on page 1? What does it mean by "distribute the optimization load over to the critic" on page 5? What does it mean by "slower than the original inefficiencies of the actor" on page 5? What does it mean precisely for the optimization landscape of Q to be more tractable? If the number of local optima of Q's landscape can be reduced, to which extent can such reduction be actually achieved?

The English presentation of this paper may need to be improved. The authors are highly recommended to conduct more rounds of proof-reading of their paper to substantially improve the presentation clarity and quality.

======

Thank the authors for responding to my questions. the response has addressed some of my concerns. Meanwhile, I have some further doubts regarding the response:

1. I understand that the new algorithm does not follow the conventional ensemble strategies. However, for the sake of showing the true strength of the new algorithm, its benefits in terms of exploration, policy gradient estimation, value function learning etc. may need to be further expanded, compared to previously proposed ensemble methods.

2. Line 13 of Algorithm 1 is based on the estimated Q-function. Due to the estimation noise/error, I don't believe it is guaranteed that the new action selection method is definitely better than the case of using a single actor. To a large extent, I feel the intuitive discussion (including some illustrative figures) about the stability issue or the direct advantage of identifying the best possible action is not sufficiently convincing. It would be great if the authors can present a more thorough theoretical analysis to justify these major claims.

3. I am not so sure how the policy gradient should be calculated for the new form of actor-critics. Specifically, does Q_i in the new form satisfy any Bellman equation and why? Why can the effective state space be expanded with actions chosen by other actors? What is the new definition of policy gradient on the basis of the expanded state space, especially when the action component of the state space is affected by the learning process of other actors (hence the expanded state space may no longer satisfy the Markov property)? Perhaps more detailed mathematical derivation could help to clarify these concerns.

**Questions:**

Why will successively restricting the action space based on the trained critic not affect the learning stability and improve the exploration effectiveness?

Why is the new policy gradient formula valid, i.e. what kind of gradient is being calculated and why the gradient allows the trained policy network to maximize the expected return?

Please refer to the previous section regarding questions on the clarity of some statements.

---

> ### Author Response · Authors · 2023-11-23
> **Thank you for your review! Here are our responses.**
>
> We express gratitude for your valuable feedback and constructive input. Your positive remarks regarding the approach and experimental performance are acknowledged and appreciated. Below, we provide responses to the concerns you raised.
>
> ### New Ensemble Baseline
> Novelty over ensemble methods: Ours is not a typical ensemble method, where usually the constituents of the ensemble are symmetric. In fact, ours is a sequential ensemble, where successive actor-critic pairs optimize different optimization problems, building up on the past actor-critic pairs.
> Our added visual result in `Fig.3` shows how this successive optimization eases the Q-function optimization landscape, which indeed helps the successive actors to find more globally optimal actions than before, with respect to the current Q-function
>
> ### Visual demonstration of improving optimization landscape
> - `[New Figure.3]` The optimization landscape of Q-value concretely represents the landscape where the N-D action space represents the independent variable and the 1-D Q-value represents the dependent variable. For visualization, we map the action space to 2-D when N > 1. Fig .3 demonstrates that the Q-function becomes more easy to globally optmize with every iteration of our successive actor-critics. This is because the locally optimal peaks below past actions’ Q-values are pruned away from the optimization landscape. As we can observe:
> - The number of local optima are reduced in $Q_0 \to Q_1 \to Q_2$, as most action values are shifted up in the Q-value space. This enables successive actors $\pi_0 \to \pi_1 \to \pi_2$ to focus on more optimal regions of the true Q-value space.
> The actions $a_1$ and $a_2$ found by $\pi_1$ and $\pi_2$ are indeed more optimal than $\pi_0$ whose actions were only locally optimal peaks, but globally suboptimal.
>
>
> ### Learning stability
> `Action selection decision`
> This is true for all actor-critic methods, especially those learned with deep learning. However, the key goal of any RL agent is to take actions that maximize its expected return in the environment, which is why actor-critic algorithms learn actors that can find actions to maximize the Q-values. This does not create any *bias* in learning of the actors, because their true objective is to maximize the critic. While the Q-function itself can be biased during deep RL training, it is still the best estimate of the objective for the actor. Our method does not degrade the primary critic’s function which is already based on stabilizing tricks adopted from TD3, like twin critics, delayed updates, and soft-action evaluation. Thus, stability of training of the primary critic is not negatively affected at all by our method. And the actor learned by our method is guaranteed to be better than a single actor, because of the max operator in Line 13 of Algorithm 1, which ensures the action is always at least as good as the one suggested by a single actor.
> Please also refer to combined response for more details.
>
> `Exploration`
> - Exploration for each successive actor $\pi_i$ still follows the standard OU Noise based exploration from DDPG, TD3. The successive actors provide an action that has a higher Q-value than a single actor would have provided, but the exploration term is still added over the actions while acting in the environment. We have modified the Algorithm 1 to clarify this important detail.
> - Furthermore, the sample efficiency improvements across 14 tasks (Fig.4 using RLiable as AC suggested) empirically justifies that our method never hurts exploration in the environment.
>
>
> ### Validity of policy gradient objective
> Each successive actor-critic pair solves its own deterministic policy gradient objective, where only the effective state space is modified. Conventional actor-critics take the form $\pi(s, a)$ and $Q(s, a)$. Whereas, each of our successive actor-critics take the form $\pi_i(\{s, a_{0:i-1}\}, a)$ and $Q_i(\{s, a_{0:i-1}\}, a, a)$. Thus, only the effective state space of the MDP is modified, and we can directly apply any actor-critic algorithm to train these actor-critics (we use TD3).
>
> Please do let us know if you have any leftover concerns or questions.

---

> ### Author Response · Authors · 2023-11-23
> **Respnose continued**
>
> ### Technical explanation and analysis of FiLM, DeepSet
> - In Appendix C.3 and C.4, we added the descriptions of Deepset along with other design choices of list-summarizers as well as Feature-wise Linear Modulation (FiLM). In short, Deepset effectively aggregates the information in the list input while FiLM helps to condition on preceding list-input. Those modules contribute to better training the list of actor-critics in SAVO.
> - We also added ablations in Appendix.D.2 & D.3 to compare (i) FiLM v/s no-FiLM and (ii) the choice of DeepSet v/s LSTM v/s Transformers to aggregate the information from previous selected actions. The results validated our choice of DeepSet as it is more scalable to an increasing number of actor-critic pairs. Also, we acknowledge that this design choice depends on the environment and the use-case as some tasks in continuous domains show that LSTM / Transformers outperform Deepset.
>
>
> ### Clarifications
> `Baseline Selection of TD3`
> We choose TD3 because of its unified applicability to both action-representation-based discrete action spaces and continuous action space. Particularly, the Wolpertinger algorithm [Dulac et al 2015] is based on the DDPG algorithm and is the most commonly used base algorithm for large discrete action spaces. While we agree with the reviewer’s suggestion that combining SAVO with SAC would be good to show wider applicability of our method, we think this extension might offer more flexibility in the design choices. Thus, for this project, we decided to stick to the simple architecture of TD3 as described above.
>
> `How to find the optimal action a' based on the primary critic`
> We improved the clarity of writing in the approach section and added the following qualitative result to support our claim;
> ``[New Figure.3]`` The optimization landscape of Q-value concretely represents the landscape where the N-D action space represents the independent variable and the 1-D Q-value represents the dependent variable. For visualization, we map the action space to 2-D when N > 1. Fig.3 demonstrates that the Q-function becomes more easy to globally optmize with every iteration of our successive actor-critics. This is because the locally optimal peaks below past actions’ Q-values are pruned away from the optimization landscape. As we can observe:
> The number of local optima are reduced in $Q_0 \to Q_1 \to Q_2$, as most action values are shifted up in the Q-value space. This enables successive actors $\pi_0 \to \pi_1 \to \pi_2$ to focus on more optimal regions of the true Q-value space.
> The actions $a_1$ and $a_2$ found by $\pi_1$ and $\pi_2$ are indeed more optimal than $\pi_0$ whose actions were only locally optimal peaks, but globally suboptimal.
>
>
> `Language clarifications`
> As per the reviewer’s suggestion, we have improved the language-ambiguous stataments:
> "navigate the action-value landscape more proficiently" → “globally optimize the Q-value landscape”
> An actor is trained with gradient ascent over the Q-value landscape and can get stuck in local optima, which we resolve.
> "distribute the optimization load over to the critic" → “utilize the critic to evaluate multiple actions”.
> "This can make the learning procedure even slower than the original inefficiencies of the actor" → “A larger action space in the joint-actor-critic model present another optimization challenge, making this solution infeasible”.

---

### Official Review · Reviewer_MBmv · 2023-10-30

**Soundness:** 1 poor
**Presentation:** 1 poor
**Contribution:** 2 fair
**Rating:** 3
**Confidence:** 5

**Summary:**

The paper proposes a continuous actor value optimization method to address the issue that traditional single actor-critic algorithms are prone to failing into local optima, in order to improve sampling efficiency and final performance in large discrete action spaces and continuous action spaces. The effectiveness of the proposed method is ultimately demonstrated through experiments.

**Strengths:**

This paper presents a novel reformulation of the actor. I think that addressing the challenge of better maximization value function by "pruning" an optimization landscape is an interesting work.

**Weaknesses:**

1) The writing expression is not sufficiently clear and the logic is confusing, especially in Introduction and Related Work sections. It is difficult to understand the structure of the article. The contribution is not clear, and it is not suitable to use a large space to introduce the experimental environment.
2) This paper lacks many vital technical explanations, including an introduction to deep-set and FiLM layer, the motivation behind their usage, and analysis of their effects.
3) The experimental results are not sufficiently reliable. The baselines are outdated. There is no mention of hyperparameter sensitivity or setting experiments.

**Questions:**

1) In Relate Work, the introduction of prior work is outdated. Please supplement it with the latest relevant work.
2) The work presented in this paper seems to fall under the domain of ensemble methods. It may be necessary to supplement it with relevant work and introduce ensemble-based value optimization algorithms as additional baselines.
3) In Algorithm 1, the "state s" in lines 8 and 10 need clarification.
4) Does the proposed method in this paper suffer from the problem of action values overestimation?
5) Why only select 3 seeds in some experiments, such as Figure 4?
6) Is the modification of the experimental setup fair? Other baselines may not have been designed specifically to address the problem presented in this paper.
7) The experiment in the Appendix only has the Easy environments, what about the other Hard environments?
8) Due to the utilization of ensemble methods, I am concerned about the computational efficiency of the algorithm. Please supplement the experiments or provide an analysis.

---

> ### Author Response · Authors · 2023-11-23
> **Added several new experiments and clarifications**
>
> We thank you for your constructive feedback that helped improve our paper significantly. We address your concerns below:
> ## Technical explanation and analysis of FiLM, DeepSet
> Please refer to the combined response
> ## Added latest baselines
> Thanks for your suggestion. Please refer to the combined response. We have added 4 new baselines: CEM, Greedy-AC, Greedy-AC-TD3, Ensemble.
> ## Added more experimental benchmarks
> > “The experimental results are not sufficiently reliable.” "The experiment in the Appendix only has the Easy environments, what about the other Hard environments?”
>
> `[Figure 2]` We added more continuous control environments in the paper, and report results across 11 environments.
> `[Figure 4(c)]` We also report the results using the RLiable library (also suggested by AC) to demonstrate that our method outperforms the baselines and ablations reliably across the environments.
> ## Computational Efficiency
> “I am concerned about the computational efficiency of the algorithm. Please supplement the experiments or provide an analysis.”
> `[Figure 4 (d)]` Thank you for this suggestion. We have added a complexity v/s performance analysis that shows the computational requirement increase is negligible. Please refer to the combined response.
> ## Supplemented discussion in Related Work
> `[Section 2]` Added discussion on evolutionary methods and ensemble methods, and also restructured the related work section significantly.
> ## Clarifications
> > “There is no mention of hyperparameter sensitivity or setting experiments.”
> - Appendix E.2 and E.3 already provides a discussion of hyperparameter sensitivity and setting details across all baselines.
> - To reiterate, the sensitive hyperparameters were found to be learning rates and network sizes of actors and critics, and were searched for each baseline.
> - `[Figure 4 (c)]` A key hyperparameter of our approach is $K$, the number of actor-critic pairs. Figure.4 shows the improvement in performance as $K$ increases in continuous control tasks. However, we chose $K$=3 across environments as that is enough to show significant gain with our method.
> - `[Appendix D2, D3]` We also add experiments to show different design choices are viable: FiLM v/s No-FiLM for conditioning, and DeepSet v/s LSTM v/s Transformer for past action summarization.
>
> > “In Algorithm 1, the "state s" in lines 8 and 10 need clarification.”
> - We correct the typo in lines 8-10, as $s$ → $s_t$. $s_t$ is the state observed in Line 5.
>
> > "Does the proposed method in this paper suffer from the problem of action values overestimation?”
> - Since our algorithm is based on TD3, the problem of overestimation is reduced by the presence of twin critics and delayed updates. We clarify this in Algorithm 1 to alleviate the concern of overstimation. To further justify, in the Q-learning update of primary critic in Line 19 of Algorithm 1, our method only modifies the search of the Q-value maximizing action. This is a key requirement of the DPG algorithm and the training objective of DPG, DDPG, TD3, etc. Works like DDPG and TD3 alleviate the issues of overestimation of Q-functions, and our implementation being based on TD3, also benefits from those solutions. All the baselines and ablations also equivalently benefit from this, and do not suffer from overestimation any more than TD3.
>
> > “Why only select 3 seeds in some experiments, such as Figure 4?”
> - All analysis results have been updated with 5 seeds. No trend is affected.
>
> > “Is the modification of the experimental setup fair? Other baselines may not have been designed specifically to address the problem presented in this paper.
> - To exemplify the problem of challenging Q-function landscapes, we curate hard benchmarks of continuous control tasks, but we also reported results on the original easy benchmarks with relatively simple optimization landscapes, in Figure 2. SAVO outperforms baselines in both the easy and hard versions of the tasks, but more significantly in the hard versions. This shows that our algorithm’s contribution is more robust to challenging optimization landscapes, that would be more common as RL scales up to harder problems. This is already evident in our results in the discrete action space environments, where the optimization landscape is already challenging because of the presence of only few valid points in the action representation space, that correspond to actual discrete actions.
>
> - The baselines from various prior works (Figure 2) are indeed incapable of addressing the problems of Q-function optimization presented in this paper. That is the key claim that we are making. All the implementations of the baselines are fair and based on the same code skeleton (code is attached in supplementary), and we clearly demonstrate that it is because of the better optimization of the Q-function by SAVO’s successive actors than baseline actors like single actors, Sampling-augmented actors, CEM Actors, and Ensemble of Actors.

---

### Official Review · Reviewer_ZeCv · 2023-10-31

**Soundness:** 2 fair
**Presentation:** 3 good
**Contribution:** 2 fair
**Rating:** 3
**Confidence:** 3

**Summary:**

The paper claims that actor often finds actions that cannot maximize the critic and this leads to sample inefficient training and suboptimal convergence. The paper proposes an algorithm that roughly works as follows. First, in addition to a primary critic, another K actors and critics are initialized, and they are queried in order: an actor’s input depends on all previous actors’ outputs; second, the action with highest primary critic value is executed; third, K updates applied to the K actor-critic pairs; last, the primary critic is updated by its own maximum action. Experiments are conducted to verify their claims.

**Strengths:**

1. The presentation is reasonably clear.

2. The proposed problem regarding the actor often cannot align well with the maximum action worths studying, it looks interesting to me.

**Weaknesses:**

The critical claims are not well-supported: 1) why the proposed method can help find maximum action; 2) the connection between finding maximum action and improved sample efficiency; 3) the actual benefit of the proposed algorithm, is it from finding the maximum, or ensemble, or exploration? 4) experiments are not well-designed; 4) highly related works are missing.

To support the claim of the paper, the following experiments need to be done:
1. add experiments to verify the proposed method does find action with a higher action value; current version directly jumps into sample efficiency, leaving the critical claim unverified;
2. The connection between finding the maximum action and improved sample efficiency is not supported, please justify;  would it introduce overestimation that hurts learning?
3. please add ensemble-based exploration method for comparison, as it is known that ensemble would provide benefits of enhancing sample efficiency. Another purpose of adding ensemble is to verify if the main benefit really comes from finding the maximum action or from exploration, If it is the letter, then the pitch of the paper should be modified and another set of baselines aiming at better exploration should be compared.
4. Any comments on the convergence of such an algorithm? I am a bit concerned that the update of an actor depends on all previous actors output could result in high non-stationarity. This would make the training difficult.
5. The proposed algorithm seems to have much higher computation cost, which weaken the practical utility.

Potential flaws of the experiment design.
1. In the algorithm, it seems at each time step, the algorithm update both policy and critic parameters K times, do the authors do the same thing for baselines?
2. Please add baselines as suggested by below missing related works.

There are several missing references that are highly relevant:

1. A model reference adaptive search method for global optimization by Jiaqiao et al.
2.  Q-learning for continuous actions with cross-entropy guided policies by Riley et al.
3.  Greedy Actor-Critic: A New Conditional Cross-Entropy Method for Policy Improvement by Samuel et al
4. CEM-RL: Combining evolutionary and gradient-based methods for policy search by Alois et al.
5. Wire fitting algorithm by Baird et al. the title is likely RL with high-dimensional continuous actions.

Among these, 2,3,4 are highly relevant and should be also compared. Please explain what the differences are between your work and those existing ones and comment on the significance of such difference. I consider this one of the critical weaknesses of this work.

**Questions:**

see above.

---

> ### Author Response · Authors · 2023-11-23
> **Important Baselines from Related Work added**
>
> We thank you for your constructive feedback that helped improve our paper significantly. We address your concerns below:
> ## Important Baselines from Related Work added:
> We appreciate the reviewer’s suggestions and pointing out the relevant work and baselines from prior work. We have added a discussion and comparison to all the suggested baselines:
>
> ### Ensemble actors
> [`Figure 2`, `Section 2.3`] As per your suggestion, we implement an ensemble-actors approach where we train multiple actors with the same critic, but explore using the critic maximizing action found by the actors. This approach helps disentangle SAVO’s ability to prune the optimization landscape from the improved exploration due to the presence of multiple actors. Please refer to combined response for further details, and Section 2.3 for a discussion and references of past work.
>
> ### Evolutionary Actors
> [`Figure 2`, `Section 2.4`] As per your suggestion, we add:
> - **CEM-actor baseline**: Algorithms like QT-Opt, CGP, CEM-RL, and GRAC employ CEM as the actor in online RL training. We implement QT-Opt style CEM, where the critic is still trained with our TD3-augmented tricks. These algorithms involving CEM require infeasible amounts of repetitive evaluations of the Q-function and do not scale well to high-dimensional action spaces (Yan et al., 2019). Note that CGP, CEM-RL and GRAC can be considered as suboptimal versions of a good CEM actor in the QT-Opt, because they already use CEM as a guide to train their actors. Thus, we only implement and compare against the QT-Opt style CEM baseline.
> - **Greedy-actor** critic baseline: emulates CEM by sampling from a high-entropy action proposal policy, evaluating these actions with the Q-function, and training the actor to greedily follow the best actions. However, since this approach also depends on gradient ascent and samples actions around the mean actions of a single-actor, thus limiting its ability to find the globally optimal action.
> - **Greedy-TD3**: We implement another version of Greedy-actor-critic in TD3 style training, where the critic now benefits from TD3 update tricks, and the exploration is performed with OUNoise added to the mean action of the stochastic greedy policy.
>
> ## Computational Cost Analysis
> `[Figure 4(d)]` Please refer to combined response.
>
> ## Clarifications
>
> > “The connection between finding the maximum action and improved sample efficiency is not supported”
> - We bring attention to Figure 4(a) and Figure(b) where we already show that SAVO results in an improved Q-value of its actions, which translates into a sample efficiency improvement when evaluating SAVO with all its actors v/s only a single actor.
>
> > “Would it introduce overestimation?”
> - No. SAVO only improves the actor’s capability to find the Q-value maximizing action, which is the key goal of the base algorithm of DPG / DDPG / TD3.
> - There is still exploration on top of the action found by SAVO, so the exploration is not hindered.
>
>
>
> > “It seems at each time step, the algorithm update both policy and critic parameters K times?”
> - No. We clarify that, as shown in Lines 16 and 17 of Algorithm 1, each individual $Q_i$ and $\pi_i$ is only updated once every iteration. $K$ denotes the number of actor-critic pairs, as written in the first line of Algorithm 1. So, the for loop in line 15 is to denote all the actor-critic pairs. To further clarify, all the baselines and ablations are based on the same algorithm design and implemented in the same code file, for fairness.
>
> ### [References]
> - Yan et al. Learning probabilistic multi-modal actor models for vision-based robotic grasping. ICRA 2019.

---

### Comment · Area_Chair_id9y · 2023-11-15
**Comment from AC**

Dear Authors,

To facilitate discussion about your paper, it would be great if you addressed the points brought up by the reviewers.

In addition, I have the following questions:
1. Towards the bottom of page 3, you introduce a nearest-neighbour mapping from continuous to discrete actions. However, if you use this mapping, the true critic will be piecewise constant, making the use of gradient based algorithms challenging. Can you elaborate exactly why this works? As I understand this, the algorithms might still work due to function approximation - although we will never learn the true (piecewise flat) critic, we will learn some version of it "smoothed out" with function approximation.
2. You base your algorithm on TD3, which uses two copies of the critic do battle overestimation bias. However, such two-critic mechanism does not seem to be included in Algorithm 1. Can you elaborate?
3. You are making claims that the critic approximation scheme you use helps battle local optima. Do you have any direct evidence supporting that hypothesis? How do you count the local optima?
4. Why not use RLiable (https://github.com/google-research/rliable) or a similar library to summarise results across environments in a robust way?

Minor nitpick: in the beginning of section 3.2, high-stake => "high-stakes".

---

> ### Author Response · Authors · 2023-11-23
> **Added visual evidence and RLiable results**
>
> Thank you for your insightful questions. We have incorporated your suggestions in the revision, and clarify your questions below:
>
> ----
> ### 1. How nearest-neighbor mapping works in the discrete action setting?
> A common strategy used in large discrete action spaces is to represent discrete actions with continuous representations, and solve the task like continuous action space RL. We follow the strategy of Wolpertinger [1], where a Q-function is learned over the continuous space. The actor’s continuous output is used to obtain k nearest-neighbor true action representations. Finally, the Q-value is evaluated over these “true” actions and the best Q-valued action is chosen.
>
> However, as you correctly point out, while training the actor with gradient ascent over the Q-function, there will be times when the critic is queried for “fake” action representations, not associated with any true discrete action. The critic would be undefined on such intermediate actions — a neural network would at-best interpolate or smooth the Q-values when queried on such samples. This creates a smoothed piecewise-like critic with a challenging optimization landscape with many locally optima, as there are peaks on the true action representations, and the in-between landscape is hard to optimize.
>
> [1] runs gradient ascent on this critic landscape completely ignoring the problem and thus underperforms empirically (Figure 2). In contrast, SAVO learns a sequence of actor-critics that successively simplify Q-value landscapes. This results in finding the action that better finds the optimal “true” peak of the Q-function.
>
> ----
> ### 2. Twin-critics of TD3 in Algorithm 1
> `[Algorithm 1]` We follow TD3 and do use twin critics for primary critic to avoid overestimation (along with other improvements like clipped double Q-learning, delayed policy updates, and target policy smoothing). As per your suggestion, we modify Algorithm 1 to denote that the update follows TD3.
>
> ----
> ### 3. Direct evidence showing reduced local optima
> `[Figure 3, 11]` As per your suggestion, we visualize the Q-value landscapes resulting in the successive critics of SAVO for all continuous control environments. For high-dimensional action spaces, we project the action space to 2D and generate a plot of Q-value over the action space. We observe:
> - successive critics $Q_0 \to Q_1 \to Q_2$ have optimization landscapes with fewer local optima, as most optima below previously selected actions are pruned away. This is visually seen as the warm (high Q-value) regions of the contour plot increase successively.
> - Better Q-valued actions are found than just a single actor $\pi_0$ that maximizes $Q_0$. Since all the peaks below the Q-value of previous actions vanish in successive critics, successive actors can focus on optimizing over better peaks.
>
> Figure 4 (a) also demonstrates our actions have 40-50% better Q-values than the baseline over the course of training, and result in better returns than if we evaluated just using a single actor.
>
> ----
> ### 4. RLiable to summarize results
> `[Figure 4 (c)]` As per your recommendation, we provide a summarized comparison across all 14 tasks with RLiable, demonstrating SAVO outperforming the baselines.
>
> ### [References]
> [1] Dulac-Arnold, Gabriel, et al. "Deep reinforcement learning in large discrete action spaces." arXiv preprint arXiv:1512.07679 (2015).

---

### Author Response · Authors · 2023-11-23
**Added: visual evidence of hypothesis, ensemble and CEM baselines, design choice validation, complexity analysis, RLiable aggregation**

We thank the reviewers for their constructive feedback that helped improve our paper significantly. All reviewers appreciated (i) the importance of the **problem** of actors being unable to find the value-maximizing action, (ii) our novel **algorithm** to prune the optimization landscape successively, and (iii) promising **experimental** results. We accommodate all feedback in the form of baselines, ablations, and visual analysis:

----
## Visual Evidence of Pruning Optimization Landscape [oxP1, AC]
`[Figure 3, 11]` We show that SAVO successfully prunes the Q-optimization landscape enabling the actors to find better globally optimal actions. Every $Q_i$ function reduces the number of locally optimal peaks, by pruning away the Q-landscape below previous actions $a_0, ..., a_{i-1}$, as seen in Fig 3 (top) results of Hopper. To obtain this result, we project the N-dimensional action space of environments to 2-D for visualization. As we see in Inverted-Pendulum-Hard result, the primary actor $\pi_0$ was stuck in a locally optimal peak. But, SAVO’s successive actor-critics finds a more optimal peak of $a_1$.

----
## New Ensemble Baseline [ZeCv, MBmv, oxP1]
`[Figure 2]` We appreciate the reviewers’ suggestion that a comparison to ensemble-based methods is necessary. We add an ensemble baseline in Fig. 3, where we train an ensemble of TD3 actors with a single primary critic. Just like our method, we select the action from the ensemble that maximizes the Q-value. Fig. 3 shows SAVO (Ours) consistently outperforms the Ensemble baseline. This helps delineate the contributions of better exploration with multiple actors with better optimization landscapes obtained using SAVO.

----
## Three New Evolution-based & CEM Baselines [ZeCv, MBmv]
`[Figure 2]` We thank the reviewers for pointing out relevant literature on evolutionary methods for better Q-function optimization by building up on CEM or emulating CEM. We add a thorough discussion in Section 2 and experiments in Section 6 on the following baselines:
- CEM-Actors: following evolutionary approaches like QT-Opt and an upper-bound to CEM-RL, CGP, GRAC (Section 2.4).
- Greedy Actor-Critic: Follows Neumann et al. (2023) by training a high-entropy proposal policy to sample actions, rate them with Q-value, and greedily train actor on best actions.
- Greedy TD3: We implement a version of Greedy-AC with our TD3-based improvements to exploration and update procedure.
SAVO outperforms all the baselines consistently, showing the importance of directly tackling the problem of challenging optimization landscapes.

----
## FiLM, DeepSet Design Choices [MBmv, oxP1, ojfo]
`[Appendix D2, D3]` We clarify the introduction of the architectural design choices of successive actor-critics in SAVO, and add a better description of these modules. In brief, the Deep Set is used as a summarizer to encode the previous actors’ actions into a latent. Alternate approaches like LSTM and Transformers can also be used. Subsequently, the encoded action-summary must condition the next successive actor and critic. This can be done with a FiLM scheme to modulate all the layers of the actor and critic neural networks, or simply by concatenating the summary with the state space. We add experiments with these design choices and a detailed discussion.

**Results**: We uniformly choose DeepSet as it is more scalable to an increasing number of actor-critic pairs, and FiLM for overall better results.

----
## Additional Results
### Computational Complexity Analysis [ZeCv, MBmv]
`[Figure 4(d)]` We investigate the impact of varying the number of actor-critic pairs on the performance of SAVO and the GPU compute memory cost across different Mujoco environments with $1 \to 3 \to 5$ actors-critics. We see a significant gain in performance, with only a minor increase in GPU usage, such as from $619 \to 637 \to 673$ MB for hopper task. In fact, prior approaches like TD3 often fail in complex optimization landscapes in both discrete and continuous action space environments that we have analyzed. Thus, we trade-off a miniscule amount of compute for significant improvements in performance and sample complexity.
RLiable results
### More Mujoco-Hard Environments [MBmv]
`[Figure 2]` Added: Inverted-Pendulum, Inverted-Double-Pendulum, Ant, HalfCheetah. SAVO consistently outperforms or matches the other best baselines across various environments.

### RLiable for aggregate statistics across 14 environments [AC]
`[Figure 4 (c)]` As per the recommendation, we provide a summarized comparison across all 14 tasks with RLiable, demonstrating SAVO outperforming the baselines.

Other editing comments and clarifications are addressed inline to reviews below. We hope our revision addresses all the concerns of the reviewers with: visual confirmation of our hypothesis, 4 additional baselines, FiLM and DeepSet design choice validation, computational complexity analysis, and consistent performance gain of SAVO across 14 tasks.

---

### Meta-Review · Area_Chair_id9y · 2023-12-05

**Metareview:**

The paper proposes a new scheme for maximising the critic in an actor-critic algorithm (like TD3 on DDPG). The proposed algorithm (SAVO) works by maintaining an an ensemble of critics and actors. Actions are then selected using a (heuristic) maximisation rule (see line 13 in Algorithm 1). Experiments are done extensively and include MineWorld, a recommender system simulator and continuous control tasks.

Strengths:
- using gradient based algorithms for recommender systems / in setting with huge action spaces is a valid and much needed area of research, in scope for ICLR.
- evaluation (after paper update) uses performance profiles generated with the rliable library
- paper is mostly well-written

Weaknesses:
- the heuristic maximisation rule is not justified formally. In other words, there is no proof that the output of Algorithm 1 solves any MDP given infinite data, which seems like a major concern for an RL method.
- the method claims to find better local optima, but the justification for that is purely experimental (no theory) and insufficient
- the method to embed discrete action spaces in a continuous space used by the paper leads to a piecewise-constant critic. This is at odds with the deterministic gradient theorem, which emphasises following the gradient of the critic. I understand the authors cite [1] which seems to do the same. However, the work [1] is not peer-reviewed.

[1] Dulac-Arnold, Gabriel, et al. "Deep reinforcement learning in large discrete action spaces." arXiv preprint arXiv:1512.07679 (2015).

**Justification For Why Not Higher Score:**

This is a strong reject. The weaknesses identified in the meta-review seem very serious. Any RL algorithm should compute the optimal policy given infinite data but I am not sure this is the case here.

**Justification For Why Not Lower Score:**

N/A

---

### Decision · Program_Chairs · 2024-01-16

Reject